# DASB - Discrete Audio and Speech Benchmark

**Pooneh Mousavi[1,2], Jarod Duret[3], Darius Petermann[6], Artem Ploujnikov[4,2], Luca Della Libera[1,2],
Anastasia Kuznetsova[6], Cem Subakan[5,2,1], Mirco Ravanelli[1,2,4]**

[1]Concordia University, [2]Mila-Quebec AI Institute, [3]Avignon Université, [4]Université de Montréal, [5]Université Laval,
[6]Carnegie Mellon University, [7]Microsoft, [8]Indiana University

**Reviewed on OpenReview:** `https://openreview.net/forum?id=HOyGvdOcUp`

## Abstract

Discrete audio tokens have recently gained considerable attention for their potential to
bridge audio and language processing, enabling multimodal language models that can both
generate and understand audio. However, preserving key information such as phonetic
content, speaker identity, and paralinguistic cues remains a major challenge. Identifying
the optimal tokenizer and configuration is further complicated by inconsistent evaluation
settings across existing studies. To address this, we introduce the Discrete Audio and Speech
Benchmark (DASB), a comprehensive framework for benchmarking discrete audio tokens
across speech, general audio, and music domains on a range of discriminative and generative
tasks. Our results show that discrete representations are less robust than continuous
ones and require careful tuning of factors such as model architecture, data size, learning
rate, and capacity. Semantic tokens generally outperform acoustic tokens, but a gap
remains between discrete tokens and continuous features, highlighting the need for further
research. DASB codes, evaluation setup, and leaderboards are publicly available at `https:
//poonehmousavi.github.io/DASB-website/`.

## 1 Introduction

Traditional speech and audio processing systems have long relied on handcrafted low-level features such
as *Mel-Frequency Cepstral Coefficients* and *Filterbanks* (Rabiner & Juang, 1993). Recently, self-supervised
learning (SSL) led to outstanding performance improvements by learning more complex, robust, and general
speech features through deep neural networks. Notable models include Wav2Vec2 (Baevski et al., 2020),
WavLM (Chen et al., 2022), and HuBERT (Hsu et al., 2021). In all these cases, the rich information in
speech and audio signals is encoded into a sequence of continuous vectors. Even though continuous vectors
have proven effective in capturing the complex details embedded in speech and audio, there is a growing
interest in discrete representations. Discrete audio representations, known as audio tokens, transform the
original waveform into a finite set of vectors. These tokens are derived using methods such as quantization of
self-supervised learning (SSL) models (Polyak et al., 2021; Mousavi et al., 2024; Wells et al., 2022; Chung
et al., 2021; Shi et al., 2024a), neural acoustic techniques (codecs)(Kumar et al., 2023; Défossez et al., 2023;
Xin et al., 2024; Wu et al., 2024c; Ji et al., 2024; Yang et al., 2024a), or hybrid approaches (Zhang et al.,
2024; Du et al., 2023; Défossez et al., 2024; Ye et al., 2025) which enhance codec representations by distilling
phonetic information into specific codebooks.

*What is driving the interest in audio tokens?* The growing interest in discrete audio tokens is inspired by the
success of autoregressive Large Language Models (LLMs) (Touvron et al., 2023; Chowdhery et al., 2024; Devlin
et al., 2019), which operate over discrete text. Motivated by this, researchers have explored representing
audio as sequences of discrete tokens (Mousavi et al., 2025; van den Oord et al., 2017; Wang et al., 2023b;
Kreuk et al., 2023), enabling the development of audio language models and multimodal LLMs (Gemini
Team, Google, 2023; Zhan et al., 2024). Discrete tokens simplify generation tasks by reframing them as
classification problems (Goodfellow et al., 2016) and enable efficient compression and storage. However,

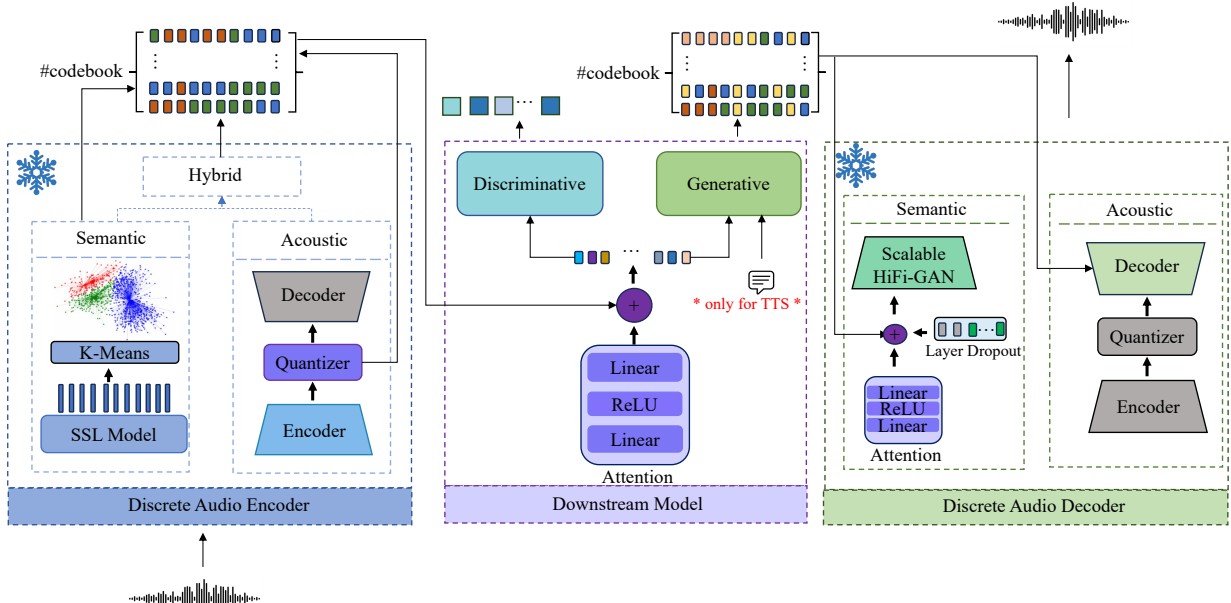

Figure 1: The workflow of DASB consists of three steps. First, a discrete audio encoder converts the audio signal into discrete tokens (*left*). Then, the tokens are combined using attention and fed to a neural model for the final prediction (*middle*). For generative tasks, the predicted tokens are passed to a discrete decoder, which converts them back into an audio waveform (*right*). Both the encoder and decoder are pretrained and frozen during downstream model training.

discretization inevitably introduces information loss. As a result, continuous features remain preferred for speech understanding tasks (Tang et al., 2024; Chu et al., 2023), while discrete tokens are increasingly favored for generation (Maimon et al., 2025; Arora et al., 2025). Ideally, audio tokens should preserve crucial information from the original waveform, including phonetic and linguistic content, speaker identity, emotion, and other paralinguistic cues. However, despite the growing interest in audio tokens (Wu et al., 2024b; Puvvada et al., 2024; Wang et al., 2024b; Zhang et al., 2023; Chang et al., 2024b; Shi et al., 2024b), there is still limited understanding of how much information is lost during tokenization across different tasks, an essential factor for building multimodal models that can both understand and generate audio. To address this gap, we introduce the **D**iscrete **A**udio and **S**peech **B**enchmark (DASB). DASB systematically assesses various audio tokens across several common audio processing tasks. In particular, our contribution is the following:

- **Direct evaluation of discrete tokens.** Unlike prior benchmarks that rely on decoding tokens back to waveforms, DASB directly evaluates tokenizers without a decoder, isolating the information loss introduced by quantization. This allows a more faithful analysis of how discrete representations impact downstream tasks and multimodal modeling. Moreover, we explicitly quantify the gap between the performance obtained with discrete representations and continous representations on a wide range of discrete and generative downstream tasks.

- **Diverse tokenizers.** We benchmark a diverse set of discrete audio encoders from all three categories: semantic (Discrete HuBERT, Discrete WavLM, Discrete Wav2Vec2) (Mousavi et al., 2024), *acoustic* (EnCodec (Défossez et al., 2023), DAC (Kumar et al., 2023), WavTokenizer (Ji et al., 2024), SQCodec (Yang et al., 2024a)), and *hybrid* (SpeechTokenizer (Zhang et al., 2024), and Mimi (Défossez et al., 2024)).

- **Comprehensive tasks and domains.** We consider a wide range of tasks, including speech recognition, speaker verification, emotion recognition, keyword spotting, intent classification, sound event classification, music genre classification, speech enhancement, speech separation, text-to-speech, and music and general sound source separation. For reliable evaluation, we apply extensive hyperparameter tuning, test two downstream architectures per task, and average results over multiple seeds.

- **Ablation studies and practical guidance.** We systematically study the effects of number of codebooks, downstream head design, and initialization strategies. These ablations provide practitioners with actionable insights into which tokenizers are best suited for specific tasks and conditions, and clarify trade-offs between reconstruction fidelity and downstream utility.

- **Public leaderboard and reproducible results.** DASB is released as a modular, reproducible benchmark built on SpeechBrain (Ravanelli et al., 2021). A public leaderboard is provided for community submissions, with code and documentation available at the official website and GitHub repository[1]

## 2 Related Work

Recent studies have explored the use of discrete audio tokens as an alternative to continuous features across a range of speech and audio tasks. Discrete representations have been applied to automatic speech recognition (Chang et al., 2023; Du et al., 2023; Mousavi et al., 2024), speech-to-speech translation (Popuri et al., 2022; Inaguma et al., 2023; Wu et al., 2023a; Chang et al., 2024a), voice conversion (Maimon & Adi, 2023; Wang et al., 2024a), text-to-speech synthesis (Ju et al., 2024; Wang et al., 2023a; Hayashi & Watanabe, 2020), speech enhancement (Wang et al., 2024b; Yang et al., 2024b; Xue et al., 2024), and source separation (Shi et al., 2021; Erdogan et al., 2023; Mousavi et al., 2024; Bie et al., 2024; Yip et al., 2024). Beyond speech, discrete tokens have also been used for music generation (Copet et al., 2023; Chen et al., 2024) and environmental sound synthesis (Yang et al., 2023b; Kreuk et al., 2023). Discrete audio tokens can be evaluated from multiple perspectives (Mousavi et al., 2025). Several benchmarks have been proposed to evaluate discrete tokenization (Shi et al., 2024a; Wu et al., 2024a; Wang & Székely, 2024; Mousavi et al., 2024). Codec-SUPERB (Wu et al., 2024a) assesses audio reconstruction quality using subjective scores and pre-trained models, while ESPnet-Codec (Shi et al., 2024a) and VERSA (Shi et al., 2025) offer a toolkit for codec training and evaluation. However, these aforementioned prior work primarily relies on reconstructed audio and focuses on a single category of tokenizers, semantic[2] or acoustic tokens.

To address the limitations of existing work, we introduce DASB, the first benchmark specifically designed to directly evaluate discrete audio tokens. Unlike prior benchmarks such as Codec-SUPER or VERSA, which decode tokens back into waveforms before evaluation, DASB evaluates tokenizers without relying on reconstruction. By removing the decoder, DASB isolates the information loss introduced by tokenization itself, providing a clearer picture of how discrete representations affect downstream tasks and multimodal language model training.However, we would like to clarify that while generative tasks require waveform reconstruction, all modeling and evaluation are performed in the discrete token space. A pre-trained frozen decoder is used only at the final stage to convert tokens into waveforms and is not part of the downstream modeling. . This distinction is critical. Reconstruction-based benchmarks can overestimate representation quality because a strong decoder may mask deficiencies in the tokens, especially since most training losses are applied to the decoder output. In contrast, DASB quantifies the true information loss from quantization and highlights trade-offs between reconstruction fidelity and downstream utility (Table 13, Appendix F). For example, tokenizers optimized for waveform fidelity often fail to preserve phonetic or semantic cues, which are essential for tasks where no decoder is involved (e.g., ASR or SLU).

Beyond isolating information loss, DASB provides a systematic framework to compare discrete and continuous representations. This is particularly relevant for multimodal models, where the choice of representation has practical consequences. Prior work shows that models such as SALMON (Tang et al., 2024) and Qwen (Chu et al., 2023) often prefer continuous features for tasks requiring deep speech understanding, while generative models tend to favor discrete tokens because classification-based training is easier than regression. DASB makes these trade-offs explicit by quantifying how much information discrete tokens retain and how this impacts task-level performance.

DASB also distinguishes itself through broad coverage. We evaluate a diverse set of tokenizers—including semantic, acoustic, and hybrid models, across both discriminative and generative tasks in speech, general

---

[1]We will release the code and evaluation pipeline with the camera-ready version.

[2]In speech processing, the term "semantic" does not align with its standard linguistic meaning. Semantic tokens in this context are better described as phonetic units (Sicherman & Adi, 2023), as they generally lack semantic content (Arora et al., 2025). For consistency across speech, audio, and music domains, we adopt the term "semantic" throughout this paper.

Table 1: Key Features of the Discrete Audio Encoders.

| Model | Abr. | SR | FR | #vocab | Domain | | | Quantization Method |
| | | | | | Speech | Music | Sound | |
|---|---|---|---|---|---|---|---|---|
| EnCodec (Défossez et al., 2023) | Enc-SMA-24 | 24KHz | 75 | 1024 | ✓ | ✓ | ✓ | acoustic - RVQ |
| DAC (Kumar et al., 2023) | DAC-SMA-24 | 24KHz | 75 | 1024 | ✓ | ✓ | ✓ | acoustic - RVQ |
| SpeechTokenizer (Zhang et al., 2024) | ST-S-16 | 16KHz | 50 | 1024 | ✓ | | | hybrid - RVQ |
| Mimi (Défossez et al., 2024) | Mimi-S-24 | 24KHz | 12.5 | 2048 | ✓ | | | hybrid - RVQ |
| Discrete WavLM (Mousavi et al., 2024) | DWavL-S-16 | 16KHz | 50 | 1000 | ✓ | | | semantic - kmeans |
| Discrete HuBERT (Mousavi et al., 2024) | DHuBERT-S-16 | 16KHz | 50 | 1000 | ✓ | | | semantic - kmeans |
| Discrete Wav2Vec2 (Mousavi et al., 2024) | DWav2Vec2-S-16 | 16KHz | 50 | 1000 | ✓ | | | semantic - kmeans |
| SQ-Codec (Yang et al., 2024a) | SQ-SMA-16 | 16KHz | 50 | 20000 | ✓ | | | acoustic - FSQ |
| WavTokenizer (Ji et al., 2024) | WT-SMA-24-2 | 24KHz | 75 | 4096 | ✓ | ✓ | ✓ | acoustic - SVQ |

audio, and music. Prior benchmarks typically focus on a narrow domain or a single type of tokenizer. In contrast, DASB adopts the design philosophy of continuous benchmarks such as SUPERB (wen Yang et al., 2021), HEAR (Turian et al., 2022), and SpeechBench (Zaiem et al., 2023), but adapts it to the unique challenges of discrete representations, which are far more sensitive to hyperparameters and architectural design choices. To ensure reliable comparisons, we perform extensive hyperparameter tuning, multi-seed evaluations, and analyze factors such as codebook size and downstream head design.

Finally, DASB provides practical guidance for researchers and practitioners. Our analyses highlight which tokenizers are most effective for different tasks, under what conditions they succeed or fail, and how design choices such as the number of codebooks or initialization strategies influence performance. Therefore, DASB complements prior works and provides practical guidance on tokenizer design and task suitability, filling crucial gaps in evaluation methodology and broadening our understanding of discrete tokenization (Mousavi et al., 2025).

## 3 Benchmark Design

The pipeline of DASB, illustrated in Figure 1, consists of three components: Audio Encoder, Downstream Model, and Audio Decoder. The following subsections describe each module in detail.

### 3.1 Tokenizer Selection Criteria

The main features of the considered tokenizers are summarized in Table 1, while Figure 2 reports their time and memory requirements for both encoders and decoders. Following the terminology from (Borsos et al., 2023; Zhang et al., 2024; Guo et al., 2025), we categorize audio tokens into three classes: semantic, acoustic, and hybrid. To ensure broad coverage, we select tokenizers that reflect diverse design choices within these categories. When available, we prioritize models trained on multiple domains (speech, music, and general audio) to maintain consistency across tasks.

**Semantic** tokens (Polyak et al., 2021; Wells et al., 2022; Chung et al., 2021) are generated by clustering or quantizing layers from SSL models (Baevski et al., 2020; Chen et al., 2022; Hsu et al., 2021). The tokenization process typically involves selecting specific layers from a pretrained SSL model and applying the k-means algorithm to group their representations. Semantic tokens primarily capture high-level information, such as phonetic, semantic, and syntactic information. They are not optimized for waveform reconstruction, making them potentially better suited for discriminative tasks like speech recognition. Recent studies, however, have shown that semantic tokens can also be effective in generative tasks (Wang et al., 2024b; Yang et al., 2024c; Mousavi et al., 2024). We adopt the tokenization algorithm proposed in (Mousavi et al., 2024). In particular, we consider three widely-used open-source SSL models: Wav2Vec2-large, WavLM-large, and HuBERT-large, each composed of 24 layers. Then, we cluster six of these layers using the k-means algorithm and select two layers from the lower part (1, 3) to capture low-level information, two from the middle layers (7, 12), and two from the higher layers (18, 23) to encode content and meaning as well.

**Acoustic** tokens (Zeghidour et al., 2021; Défossez et al., 2023; Kumar et al., 2023) are primarily designed for audio compression and are trained to accurately reconstruct the original waveform, making them suitable

for generation tasks. Our benchmark includes four acoustic tokenizers reflecting different design choices. EnCodec (Défossez et al., 2023) uses a 1D convolutional encoder, a two-layer LSTM, and a decoder, with Residual Vector Quantization (RVQ) (Zeghidour et al., 2021) applied to compress the latent space. It is trained end-to-end with a combination of reconstruction and perceptual losses and supports multiple bitrates from 1.5 to 24 kbps. DAC (Kumar et al., 2023) improves upon EnCodec by adopting advanced vector quantization techniques from the image domain, enhanced adversarial training, and quantizer dropout to enable flexible bitrate control. Beyond RVQ-based models, WavTokenizer replaces RVQ with single vector quantization (SVQ), expanding the vocabulary size to simplify the model architecture and avoid the complexity of managing multiple codebooks. Finally, SQ-Codec (Mentzer et al., 2023; Yang et al., 2024a) adopts finite scalar quantization (FSQ), mapping each feature dimension independently to a fixed set of scalar values. This approach improves codebook utilization, prevents codebook collapse, and creates a scalar latent space rather than relying on traditional vector quantization.

**Hybrid** tokenizers (Zhang et al., 2024; Du et al., 2023) combine semantic and acoustic information by disentangling different aspects of speech hierarchically across quantization layers. SpeechTokenizer (Zhang et al., 2024) and Mimi (Défossez et al., 2024) adopt this approach by introducing semantic distillation into the residual vector quantization (RVQ) process. Both models build on RVQ-GAN architectures, where a semantic teacher network supervises the first RVQ quantizer to encourage early tokens to capture phonetic and linguistic content. Subsequent RVQ layers capture residual paralinguistic features such as speaker identity and prosody, while preserving low-level acoustic details necessary for reconstruction. In SpeechTokenizer, HuBERT is used as the semantic teacher, whereas Mimi uses WavLM. The training objective includes maximizing the cosine similarity between the outputs of the first RVQ layer and the semantic teacher representations.

## 3.2 Discrete Audio Encoder

The audio encoder converts the audio signal into a sequence of discrete tokens. It is pretrained on large amounts of unlabeled data and remains frozen during the training of downstream tasks. Different encoders may compress the information in the original waveform at different rates. The compression level is measured by the bitrate, defined as:

$$\text{bitrate} = \log_2 V \cdot C \cdot R, \tag{1}$$

where $C$ is the number of codebooks, $V$ is the number of vectors in each codebook (vocabulary), and $R$ is the frame rate of codes per second. It is worth mentioning that a single sequence of tokens might be insufficient to capture the rich and complex information embedded in speech signals. The encoders thus often output multiple discrete sequences, with each sequence corresponding to a different codebook $C$. The encoders can operate at different bitrates simply by adjusting the number of codebooks $C$. However, this introduces a trade-off between efficiency and performance. To ensure a fair comparison, we evaluate tokenizer performance across multiple bitrate settings: the lowest, the highest, and the configuration recommended by the original authors. If no alternatives are provided, we use the default configuration. We consider this approach to prevent the trivial conclusion that some audio tokens perform better than others simply due to a higher bitrate.

## 3.3 Downstream Model

In this step, we employ neural networks to solve supervised tasks of common interest. We first map discrete tokens to embeddings and then dynamically combine embeddings from different codebooks using an attention mechanism. A simple multi-layer perceptron (MLP) computes attention scores across codebooks at each time step, allowing the model to adaptively weigh information based on task requirements. The MLP generates a score for each codebook, which is normalized by a softmax function, as shown below:

$$z_{c,t} = f\big(\text{emb}(d_{c,t})\big), \quad a_{c,t} = \frac{\exp(z_{c,t})}{\sum_{k=1}^{C} \exp(z_{k,t})}, \quad h_t = \sum_{c} a_{c,t} z_{c,t}, \tag{2}$$

where $d_{c,t}$ is the discrete token from codebook $c$ at time $t$, $z_{c,t}$ is the MLP score, and $\text{emb}(\cdot)$ refers to the lookup table mapping tokens to embeddings. The attention weight $a_{c,t}$ determines the contribution of each codebook, and $h_t$ is the final combined representation passed to the downstream model. All embeddings are

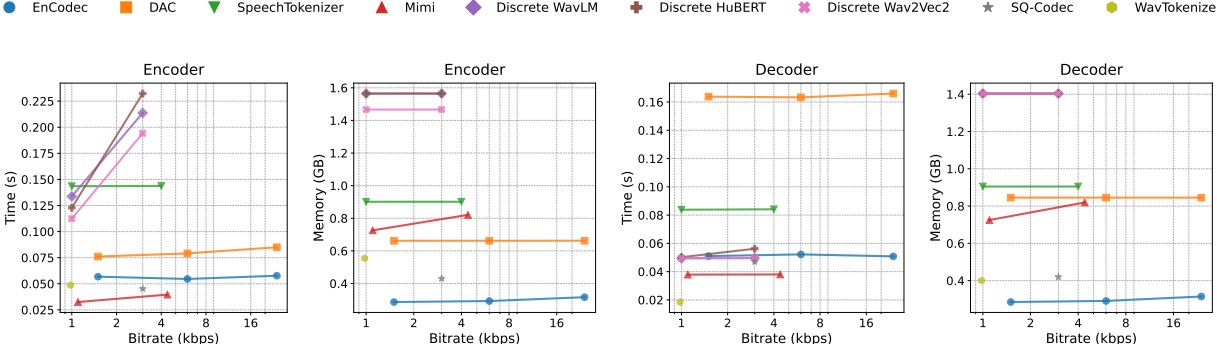

Figure 2: Time and memory required to process a 16-second utterance for the encoders and decoders of the considered audio tokenizers, measured on an NVIDIA Quadro RTX 8000 GPU with 128 GB RAM. The number of points per tokenizer depends on the available bitrate configurations.

randomly initialized with a dimension of 1024 to ensure consistency. We study the impact of embedding initialization in Appendix E. For SQ-Codec, which uses scalar quantization and Group VQ, we replace learnable embeddings with a ternary-matrix-based approach. The final 1024-dimensional embedding is constructed by concatenating four separate 256-dimensional group embeddings to efficiently handle its large vocabulary (~20k tokens). The combined representations $h_t$ are fed into task-specific neural models, trained end-to-end along with their attention and embedding layers. For discriminative tasks, the model outputs either a single or sequential prediction, while for generative tasks, it predicts token sequences per codebook, which are later decoded into audio.

### 3.4 Discrete Audio Decoder

The decoder, used for generative tasks only, converts the predicted tokens into audio signals. The decoder is frozen during training. The choice of decoder depends on the encoder used in the first step. For acoustic and hybrid tokenizers, we use their built-in decoder. For semantic tokens, we use the scalable vocoder proposed in (Mousavi et al., 2024), which is a modified HiFi-GAN (Yang et al., 2023a) pretrained with LibriSpeech-960h (Korvas et al., 2014). The scalable vocoder accepts a variable number of multi-layer semantic tokens as input and can handle different bitrates using a layer dropout mechanism.

## 4 Experiments

We evaluate each tokenizer across multiple bitrate settings and tune two key hyperparameters (number of layers and learning rate) using the TPE algorithm (Watanabe, 2023), implemented through the Orion framework (Bouthillier et al., 2021). Each configuration is tuned over 20 trials and evaluated using three random seeds. To ensure fair comparison, we adopt simple downstream architectures that highlight the quality of the tokens rather than compensating with model complexity. Our objective is not to achieve state-of-the-art results on individual tasks but to provide a fair and consistent evaluation across different tokenizers. Each task is evaluated with both shallow and deep models to study the effect of model capacity and improve robustness. For continuous baselines, we use the same downstream setup as for discrete tokens, with WavLM weighted-sum features unless otherwise specified. This design isolates the effects of discrete versus continuous representations under a shared architecture. Additional experimental details are provided in Appendix G.

### 4.1 Discriminative Tasks

- **Automatic Speech Recognition (ASR)**: We evaluate ASR on two settings: English ASR using LibriSpeech (Korvas et al., 2014) and low-resource ASR on Welsh and Basque from CommonVoice 17.0 (Ardila et al., 2020). For English ASR, training uses the train-clean-960 subset, validation is performed on dev-clean, and testing is conducted on test-clean and test-other subsets. For CommonVoice datasets,

we follow the official train/validation/test splits. We use two downstream architectures: (i) a multi-layer BiLSTM with a linear layer, and (ii) a Branchformer (Peng et al., 2022) with sequence-level cross-entropy and CTC loss. Word Error Rate (WER) is used for evaluation.

- **Speaker Identification/Verification (SID, SV)**: Speaker Identification involves classifying each utterance by its speaker identity as a multi-class classification, with the same predefined set of speakers for both training and testing. The evaluation metric is the accuracy. Automatic Speaker Verification (ASV), instead, involves training a binary classifier to determine whether the speakers in a pair of utterances are the same. The evaluation metric adopted in this case is the equal error rate (EER). We use the widely-used VoxCeleb1 (Nagrani et al., 2017) train and test splits for both tasks. First, we test the ECAPA-TDNN (Desplanques et al., 2020) architecture with AM-Softmax (Wang et al., 2018) loss for training the speaker embeddings. For verification, we use the cosine similarity between speaker representations. As a second architecture, we replace the ECAPA-TDNN with a BiLSTM (followed by a linear classifier).

- **Emotion Recognition (ER)**: The task involves predicting one of the four classes: *happy*, *sad*, *angry*, and *neutral*. We use the popular IEMOCAP (Busso et al., 2008) dataset, which contains about 10k samples from 10 speakers. As a first architecture, we use ECAPA-TDNN. For the second downstream architecture,we directly input the representations into a linear classification layer after averaging them along the time axis. The evaluation metric is the accuracy.

- **Intent Classification (IC)**: This task aims to determine the intention or purpose given utterance a speech recording. In particular, we here aim to classify each utterance into one of 18 scenarios, including *calendar*, *email*, and *alarm*. For this task, we utilize the SLURP dataset (Bastianelli et al., 2020), which comprises around 72k audio recordings of single-turn user interactions with a home assistant. We employ ECAPA-TDNN and a BiLSTM (followed by a linear classifier) as downstream architectures. We evaluate the performance using the accuracy.

- **Keyword Spotting (KS)**: Keyword Spotting involves detecting predefined keywords by classifying utterances into a set of specified words. We use the Speech Commands dataset v1.0 (Warden, 2018) for this task. The dataset includes ten classes of keywords, a class for silence, and an unknown class to account for false positives. We employ both the X-vector and ECAPA-TDNN architectures. The evaluation metric is the accuracy.

- **Event Sound Detection (ES)**: This task involves classifying audio clips into 50 different sound event categories, using the ESC-50 dataset (Piczak, 2015), which contains 2,000 labeled 5-second recordings across 50 sound classes (40 samples per class). We employed two downstream architectures: linear and ECAPA-TDNN, with accuracy as the evaluation metric. For a continuous baseline, we utilized the pre-trained Beats model(Chen et al., 2023).

- **Music Genre Classification (MG)**: This task focuses on classifying music clips into 10 genre categories using the GTZAN dataset (Tzanetakis & Cook, 2002), which contains 100 audio files per genre, each 30 seconds long. Similar to event sound detection, we used two downstream architectures: linear and ECAPA-TDNN, with accuracy as the evaluation metric. We used the MERT (Li et al., 2023) model, which has a similar architecture to wav2vec2 as a continuous baseline.

- **Urban Sound Classification (USC)**: This task involves classifying audio clips into 10 urban sound categories using the UrbanSound8K dataset (Ooi et al., 2021), which contains 8732 labeled audio clips (4 seconds) across 10 classes. We employed two downstream architectures: Xvcetor and ECAPA-TDNN, with accuracy as the evaluation metric.

- **Pitch Classification (PC)**: This task involves classifying instrumental sounds into 88 pitch classes using the NSynth dataset (Engel et al., 2017), which contains a large collection of annotated musical notes. The goal is to predict the pitch of each audio sample, and performance is measured using pitch accuracy. We employed two downstream architectures: Xvcetor and ECAPA-TDNN.

- **Music vs Speech Classification (MSC)**: This task focuses on distinguishing between music and speech using the GTZAN Music-Speech subset (Tzanetakis, 1999), which contains 120 audio tracks (60 per class), each 30 seconds long. We employed two downstream architectures: Xvcetor and ECAPA-TDNN, with accuracy as the evaluation metric.

## 4.2 Generative Tasks

- **Speech Enhancement (SE)**: Speech enhancement aims to improve audio quality by cleaning up noisy input recordings. For this task, we utilize the popular VoiceBank dataset (Valentini-Botinhao et al., 2016). We employ two downstream architectures: a non-autoregressive Conformer encoder (Gulati et al., 2020), and a convolutional recurrent deep neural network (CRDNN). The input tokens are extracted from the noisy signal, while target tokens from the clean one. Training is performed using the cross-entropy loss. The speech quality is assessed using the deep noise suppression mean opinion score (DNSMOS) (Reddy et al., 2022). The intelligibility is evaluated through the differential word error rate (dWER) (Wang et al., 2021), which measures the WER between the transcribed enhanced signal and the transcribed target signal. The transcriptions are obtained using the small version of Whisper (Radford et al., 2022). Additionally, to measure speaker fidelity, we use the cosine similarity (SpkSim) between X-vectors extracted from the enhanced signal and the target signal using the base variant of WavLM (Chen et al., 2022) fine-tuned for speaker verification.

- **Speech Separation (SS):** SS aim at isolating individual voices in multi-speaker signals using Libri2Mix (Cosentino et al., 2020). We employ a Conformer and CRDNN, trained using permutation-invariant loss (Kolbæk et al., 2017). The same evaluation metrics as SE are used. We adopt the same architecture as in the discrete experiments for the continuous baseline, using a weighted sum of WavLM as input. While models such as Conv-TasNet (Luo & Mesgarani, 2019) or Transformer-based ones (Saijo et al., 2024) are well-established baselines, we do not include them here as Libri2Mix has become a largely saturated benchmark, with many approaches already achieving near-ceiling performance. In this context, adding stronger separation backbones would not yield meaningful insights. Instead, our focus is on isolating the effects of discrete versus continuous representations under a shared architecture. For more challenging domains like general audio and music, where separation remains an open problem, we do include SOTA architectures to establish stronger baselines.

- **Music Separation:** Music source separation (MSS) aims at isolating individual instrumental components from a musical mixture. Typically, sources are categorized as "Bass", "Drums", "Vocals" and "Others" with the latter encompassing any remaining elements. Similar to our speech separation task, we employ a Conformer-based and CRDNN model trained on the MUSDB dataset (Rafii et al., 2017a). We evaluate its performance using the BSSEval toolkit (Vincent et al., 2006), which provides metrics such as signal-to-distortion ratio (SDR), signal-to-artifact ratio (SAR), and signal-to-interference ratio (SIR). As a continuous baseline, we use the music source separation model DEMUCS (Rouard et al., 2023), which achieves state-of-the-art results in music source separation.

- **Sound Separation:** For general audio source separation, we employ a Conformer-based and CRDNN model trained on the FUSS dataset (Wisdom et al., 2021). Unlike speech and music separation, where sources are well-defined (e.g., vocals, instruments, or speakers), FUSS consists of arbitrary sound mixtures, making the separation task more challenging. We evaluate performance using SDR as well. We employ the TDCN++ masking network as our continuous baseline system (Kavalerov et al., 2019).

- **Text-to-Speech (TTS)**: We evaluate both single-speaker and multi-speaker TTS tasks. For single-speaker TTS, we use a simple autoregressive encoder-decoder Transformer (Vaswani et al., 2017) trained on LJSpeech (Ito, 2017) with raw text inputs. For multi-speaker TTS, we adopt a simplified version of ESPNet's VALL-E (Wang et al., 2023a) trained on LibriTTS-1000 (Zen et al., 2019) with phoneme inputs provided by LibriTTS alignments (Minixhofer, 2023). To ensure a fair comparison, we keep all training settings, including the number of epochs and sampling strategies, consistent across tokenizers. Speech quality is assessed using the pretrained UTokyo-SaruLab System (UTMOS) (Takaaki et al., 2022). Pronunciation accuracy is measured using dWER, and speaker similarity is also reported following the

Table 2: DASB results for discriminative tasks (speech) with the first downstream architecture. #Q refers to number of codebook.

| Models\Tasks | #Q | ASR-En WER↓ | | ASR-LR WER↓ | | ER ACC↑ | IC ACC↑ | KS ACC↑ | SI ACC↑ | SV EER↓ |
|---|---|---|---|---|---|---|---|---|---|---|
| | | Clean | Other | Welsh | Basque | | | | | |
| Continuous | – | **2.26** | **5.09** | **41.77** | **14.32** | **63.10** | **86.10** | **99.00** | **99.70** | **2.10** |
| Enc-SMA-24 | 2 | 39.16±1.85 | 63.44±1.35 | 90.90±0.32 | 51.00±0.98 | 45.50±1.40 | 42.90±0.16 | 77.73±3.12 | 89.81±5.46 | 18.33±0.26 |
| | 8 | 16.83±1.32 | 38.98±2.16 | 84.53±1.90 | 45.36±0.57 | 44.73±2.23 | 40.03±0.29 | 74.30±1.69 | 94.26±3.99 | 13.54±0.57 |
| | 32 | 50.84±34.53 | 66.04±24.79 | 97.39±1.19 | 58.21±0.92 | 42.96±1.75 | 33.66±2.65 | 69.10±3.42 | 91.12±1.92 | 10.12±6.66 |
| DAC-SMA-24 | 2 | 24.57±0.16 | 50.60±0.28 | 95.21±0.84 | 68.93±0.42 | 45.20±1.50 | 29.83±0.19 | 67.27±1.56 | **97.88±0.79** | 15.86±5.26 |
| | 8 | 16.29±0.28 | 39.57±0.29 | 97.20±0.14 | 62.45±1.40 | 44.73±1.80 | 23.97±0.41 | 65.27±2.82 | 87.33±10.98 | 15.86±5.26 |
| | 32 | 98.40±0.34 | 98.42±0.32 | 98.96±0.18 | 73.57±1.56 | 43.20±2.30 | 44.60±39.19 | 68.67±2.91 | 87.69±4.99 | 17.12 ± 0.76 |
| ST-S-16 | 2 | 14.92±1.20 | 34.19±1.37 | 71.36±0.32 | 42.17±0.05 | 54.86±0.85 | 56.80±0.08 | 94.11±0.63 | 73.16±0.37 | 24.23±0.29 |
| | 8 | 12.69±0.42 | 30.99±0.65 | 68.36±0.44 | 35.35±0.22 | 55.00±0.77 | 53.83±0.05 | 94.11±0.07 | 96.78±0.45 | 10.45±0.43 |
| Mimi-S-24 | 8 | 13.96±0.16 | 31.53±0.26 | 91.59±0.15 | 59.18±8.52 | 51.13±1.57 | 53.83±0.19 | 92.18±0.20 | 79.50±0.43 | 18.68±0.35 |
| | 32 | 13.52±0.13 | 30.77±0.17 | 96.89±0.07 | 58.15±6.90 | 46.76±0.61 | 50.73±0.50 | 91.31±0.19 | 63.93±13.64 | 23.91±4.60 |
| DWavL-S-16 | 2 | 6.00±0.40 | 14.38±0.58 | 58.98±0.15 | 22.02±0.17 | 61.53±1.76 | 76.33±0.17 | **96.82±0.92** | 76.57±0.33 | 22.41±0.19 |
| | 6 | **4.32±0.23** | **10.73±0.42** | **48.94±0.38** | **19.66±0.33** | **63.20±0.85** | **78.73±0.12** | 95.89±0.50 | 92.31±0.09 | 13.47±0.22 |
| DHuBERT-S-16 | 2 | 5.89±0.19 | 14.71±0.33 | 53.51±0.73 | 22.16±0.26 | 55.97±0.63 | 67.00±1.061 | 82.9±0.09 | 86.39±10.64 | 23.15±0.07 |
| | 6 | 4.37±0.31 | 10.92±0.56 | 54.74±0.48 | 20.83±0.07 | 59.90±1.73 | 71.20±0.10 | 88.10±1.70 | 66.53±0.24 | 22.64±0.29 |
| DWav2Vec2-S-16 | 2 | 4.47±0.34 | 32.87±0.47 | 61.85±0.09 | 28.68±0.46 | 56.43±0.39 | 67.90± 0.35 | 93.45±1.80 | 76.86±8.44 | 24.49±0.43 |
| | 6 | 6.57±0.24 | 15.94±0.31 | 54.35±0.25 | 23.19±0.40 | 60.63±0.90 | 71.10±0.10 | 96.24±0.81 | 89.76±0.30 | 14.41± 0.31 |
| SQ-SMA-16 | 4 | 11.63±0.08 | 30.91±0.17 | 94.80±0.88 | 94.24±1.24 | 41.30±5.48 | 58.13±0.26 | 92.74±0.42 | 97.38±0.03 | **9.69±0.25** |
| WT-SMA-24-2 | 1 | 28.69±0.96 | 52.90±1.12 | 97.41±0.08 | 75.82±0.20 | 43.43±2.41 | 15.25±0.15 | 59.13±2.10 | 85.90±2.48 | 19.38±0.36 |

Table 3: DASB results for generative tasks (speech) with the first downstream architecture. #Q refers to number of codebook.

| Models\Tasks | #Q | SE DNSMOS↑ | dWER↓ | Spk Sim↑ | SS - Speech DNSMOS Rec↑ | DNSMOS Sep↑ | dWER↓ | Spk Sim↑ | TTS UTMOS↑ | dWER↓ | Spk Sim↑ |
|---|---|---|---|---|---|---|---|---|---|---|---|
| Continuous | – | **3.49** | **4.92** | **0.93** | – | **3.68** | **9.97** | **0.94** | **3.10** | **7.14** | **0.83** |
| Enc-SMA-24 | 2 | 3.15 ± 0.01 | 34.95 ± 0.64 | 0.86 ± 0.00 | 3.19 | 3.13±0.00 | 80.33±1.77 | 0.88±0.0 | 1.82±0.00 | 4.64±0.28 | 0.88±0.00 |
| | 8 | 3.08 ± 0.01 | 22.70 ± 1.84 | 0.88 ± 0.00 | 3.54 | 3.08±0.00 | 53.37±0.65 | 0.90±0.00 | 2.95±0.00 | 5.82±0.12 | **0.91±0.00** |
| | 32 | 2.78 ± 0.01 | 65.70 ± 6.09 | 0.80 ± 0.01 | 3.72 | 2.97±0.01 | 92.42±0.97 | 0.85±0.0 | 1.94±0.15 | 23.29±0.50 | 0.87±0.00 |
| DAC-SMA-24 | 2 | 3.26 ± 0.01 | 54.85 ± 1.82 | 0.86 ± 0.00 | 3.16 | 3.01±0.0 | 101.19±1.99 | 0.85±0.00 | 2.37±0.05 | 8.34±0.39 | 0.88±0.00 |
| | 8 | 3.51 ± 0.01 | 29.44 ± 3.93 | **0.90 ± 0.01** | 3.67 | 3.3±0.00 | 52.77±2.48 | **0.93±0.0** | 3.19±0.01 | 12.59±0.48 | 0.88±0.00 |
| | 32 | 2.93 ± 0.01 | 30.66 ± 0.97 | 0.88 ± 0.00 | **3.76** | 2.67±0.01 | 92.07±0.05 | 0.88±0.01 | 1.64±0.02 | 21.55±0.11 | 0.86±0.01 |
| ST-S-16 | 2 | 3.19 ± 0.02 | 29.98 ± 0.58 | 0.86 ± 0.00 | 3.20 | 3.13±0.0 | 84.94±0.63 | 0.87±0.00 | 1.65±0.04 | 4.31±0.77 | 0.89±0.00 |
| | 8 | 3.49 ± 0.01 | 21.65 ± 0.57 | 0.87 ± 0.00 | 3.72 | 3.43±0.01 | 60.90±0.77 | 0.91±0.00 | 3.37±0.24 | 10.39±1.60 | **0.91±0.00** |
| Mimi-S-24 | 8 | 3.25 ± 0.01 | 67.56 ± 2.21 | 0.85 ± 0.00 | 3.65 | 3.29±0.0 | 109.30±3.3 | 0.87±0.00 | 2.62±0.04 | 9.62±0.38 | **0.91±0.00** |
| | 32 | 3.18 ± 0.01 | 102.61 ± 2.40 | 0.82 ± 0.00 | 3.72 | 3.00±0.00 | 137.00±2.16 | 0.82±0.0 | 1.33±0.00 | 65.31±0.00 | 0.83±0.00 |
| DWavL-S-16 | 2 | 3.56 ± 0.01 | 25.88 ± 2.15 | 0.88 ± 0.00 | 3.57 | 3.75 | 49.57±0.64 | 0.85±0.0 | 2.90±0.01 | 3.32±0.03 | 0.86±0.00 |
| | 6 | **3.57 ± 0.01** | **9.43 ± 0.33** | 0.89 ± 0.00 | 3.75 | **3.75±0.01** | **30.39±0.45** | 0.91±0.00 | 3.79±0.01 | 3.93±0.14 | 0.90±0.00 |
| DHuBERT-S-16 | 2 | 3.45 ± 0.01 | 59.00 ± 3.92 | 0.85 ± 0.00 | 3.68 | 3.63±0.01 | 80.92±1.88 | 0.84±0.00 | 3.36±0.00 | **3.28±0.29** | 0.85±0.00 |
| | 6 | 3.49 ± 0.01 | 12.74 ± 0.15 | 0.88 ± 0.00 | 3.72 | 3.68±0.01 | 65.8±1.46 | 0.89±0.00 | **3.94±0.00** | 3.70±0.07 | 0.90±0.00 |
| DWav2Vec2-S-16 | 2 | 3.55 ± 0.0 | 30.14 ± 0.44 | 0.88 ± 0.00 | 3.61 | 3.58±0.01 | 85.15±0.2 | 0.82±0.00 | 2.82±0.02 | 5.58±0.06 | 0.82 ± 0.00 |
| | 6 | **3.57 ± 0.01** | 12.93 ± 0.56 | 0.89 ± 0.00 | 3.72 | 3.69±0.07 | 70.99±10.27 | 0.87±0.04 | 3.28±0.03 | 12.21±1.06 | 0.88±0.00 |
| SQ-SMA-16 | 4 | 3.28 ± 0.01 | 122.33 ± 8.74 | 0.83 ± 0.00 | 3.77 | 3.19±0.00 | 136.00±3.58 | 0.83±0.00 | – | – | – |
| WT-SMA-24 | 1 | 3.33 ± 0.01 | 67.53 ± 10.65 | 0.85 ± 0.00 | 3.57 | 3.42±0.0 | 118.33±4.50 | 0.86±0.0 | 2.86±0.04 | 4.95±0.00 | 0.89±0.00 |
| Mixture | – | – | – | – | – | 3.43 | – | – | – | – | – |

same setup as SE. For the multi-speaker continuous baseline, we use a Tacotron2 variant (Shen et al., 2018) pretrained on LibriTTS[3].

-

## 5 Results

Tables 2, 3, 4, and Appendix D summarize the performance on discriminative and generative tasks across the two downstream architectures evaluated (see Appendix C for the ordering of the first and second heads). We observe notable variation in tokenizer performance across different tasks and domains, suggesting that the

---

[3]https://huggingface.co/speechbrain/tts-mstacotron2-libritts

Table 4: DASB results for generative and discriminative tasks (music - audio) with the first downstream architecture. #Q refers to number of codebook.

| Models\Tasks | #Q | SS - Audio SI-SDRi↑ | | SS - Music SI-SDRi↑ | | SAR↑ | SIR↑ | ESC ACC↑ | USC ACC↑ | MSC ACC↑ | PC ACC↑ | MGC ACC↑ |
| | | Rec | Sep | Rec | Sep | | | | | | | |
|---|---|---|---|---|---|---|---|---|---|---|---|---|
| Continuous | – | – | 15.07 | – | 13.29 | 9.56 | 11.99 | 92.91 | 85.70 | 100.00 | 90.60 | 87.00 |
| Enc-SMA-24 | 2 | 0.76 | 7.03 ± 0.49 | 3.36 | 1.49 ± 2.04 | -2.80 ± 1.68 | 5.96 ± 1.52 | 34.83 ± 0.47 | 60.03±2.24 | 100.00±0.00 | 69.80±1.88 | 70.33 ± 1.70 |
| | 8 | 3.87 | 9.53 ± 0.33 | 7.99 | 1.98 ± 0.36 | -1.95 ± 0.33 | 5.26 ± 0.22 | 37.00 ± 0.73 | 59.43±1.55 | 97.22±3.93 | 65.40±1.61 | 54.67 ± 3.86 |
| | 32 | 5.76 | -1.73 ± 0.09 | 11.10 | -11.72 ± 0.35 | -15.00 ± 0.02 | -0.42 ± 0.01 | 35.43 ± 1.45 | 55.70±2.58 | 94.45±3.93 | 60.27±0.62 | 39.67 ± 1.25 |
| DAC-SMA-24 | 2 | 0.12 | 3.84 ± 0.48 | 2.37 | 1.01 ± 0.17 | -3.59 ± 0.09 | 5.92 ± 0.28 | 31.03 ± 1.84 | 46.53±3.42 | 97.22±3.93 | 50.33±1.75 | 50.00 ± 0.82 |
| | 8 | 3.33 | 5.62 ± 0.21 | 6.66 | -11.77 ± 0.1 | -10.62 ± 2.35 | -5.52 ± 3.68 | 28.60 ± 0.79 | 46.53±3.59 | 91.67±0.00 | 53.47±1.51 | 47.67 ± 3.09 |
| | 32 | 4.73 | -4.92 ± 0.32 | 8.54 | -11.32 ± 0.12 | -12.70 ± 0.17 | -2.05 ± 0.41 | 36.67 ± 0.92 | 40.00±2.26 | 94.45±3.93 | 60.27±0.62 | 50.00 ± 0.82 |
| SQ-SMA-16 | 4 | 3.62 | 6.54 ± 0.22 | 5.53 | -3.62 ± 0.87 | -5.84 ± 0.86 | 1.42 ± 0.32 | 31.37 ± 1.37 | 35.47±1.96 | 94.45±3.93 | 43.87±2.13 | 42.67 ± 0.47 |
| WT-SMA-24-2 | 1 | -24.05 | -16.72 ± 0.08 | -2.66 | -4.52 ± 0.04 | -8.32 ± 0.07 | 2.65 ± 0.11 | 34.50 ± 0.82 | 45.87±3.17 | 100.00±0.00 | 44.90±2.41 | 48.00 ± 1.41 |
| Mixture | – | – | -16.5 | – | -7.71 | 50.01 | -inf | – | – | – | – | – |

best tokenizer depends on the specific task. However, several consistent patterns emerge, which we highlight below.

## 5.1 Comparison of Discrete Audio Models

**Speech Tasks.** Semantic tokens outperform acoustic tokens in most discriminative tasks, reflecting their ability to preserve phonetic content, as also mentioned in (Borsos et al., 2023). Hybrid tokenizers based on semantic distillation rank second. Speaker recognition tasks are an exception, where DAC achieves the best results. This suggests that acoustic tokens, trained with reconstruction objectives, better capture speaker identity. This finding is consistent with (van Niekerk et al., 2022), which shows that quantizing SSL representations tends to remove speaker-specific information. A similar trend appears in generative tasks, where semantic tokens achieve the highest MOS and dWER scores, followed by hybrid tokenizers. For speaker similarity, acoustic tokenizers perform better. We also observe that, in separation tasks, reconstructed DNSMOS scores often fail to surpass the "Unprocessed" baseline, indicating that reconstruction quality can limit downstream performance, especially for tasks requiring high-fidelity signals.

**Audio and Music Tasks.** For general audio and music tasks, EnCodec consistently outperforms other tokenizers across all bitrates and domains. DAC performs worse, likely due to its emphasis on perceptual quality at the expense of signal fidelity. The difficulty of these tasks is evident from the low SI-SDR of the unprocessed mixtures, approximately $-16$ dB for general audio and $-7.7$ dB for music. Even the best-performing model (EnCodec at medium bitrate) achieves only around $-7$ dB SI-SDR for audio and $-5.7$ dB for music.

**Continuous vs. Discrete Representations.** Across all domains and tasks, continuous representations outperform discrete tokens. Tokenization causes information loss, affecting crucial aspects such as phonetics, speaker identity, and emotion. Our findings point to a critical research direction: *addressing information loss to enable the next generation of audio tokenizers suitable for training multimodal models capable of both audio understanding and generation tasks.*

## 5.2 Impact of Tokenizer Design and Evaluation Setting

**Impact of Codebook Size and Bitrate.** We study the effect of bitrate and codebook size on tokenizer performance. Tables 2, 3, and 4 show that medium bitrate settings achieve the best results across both discriminative and generative tasks. Higher bitrates, while preserving more information, often degrade performance by increasing output dimensionality and modeling complexity. Similarly, increasing the number of codebooks improves signal reconstruction but tends to reduce downstream task performance. In RVQ-based models, earlier codebooks capture more phonetic information, while later ones often add redundancy, which may explain this trade-off. This observation is also supported by prior work. Recent studies show that higher bitrates do not always lead to better downstream performance. For example, Moshi (Défossez et al., 2024) reports that lower bitrates can sometimes yield better results. Similarly, VoxtLM (Maiti et al., 2024) shows that increasing the number of centroids from 200 to 1000 worsens CER ($3.5 \rightarrow 6.1$) in TTS. These findings suggest that higher bitrate can introduce redundancy and complexity that harm semantic learning, even if reconstruction improves. This highlights an important design principle for tokenizers: *optimizing for*

*reconstruction alone does not guarantee better performance on downstream tasks.* Additional reconstruction results for each tokenizer are provided in Appendix F.

**Efficiency.** We also evaluate the computational efficiency of encoders and decoders, as this can impact real-world applications. Figure 2 shows the time and memory usage for each encoder-decoder pair across all bitrate settings. Semantic tokenizers typically use large, computationally heavy encoders, while their decoders, based on compact HiFi-GAN models, are much more efficient. For streaming applications (Wu et al., 2023b) where time and memory constraints are critical, Mimi and Encodec stand out as a better candidate due to the efficiency of both its encoder and decoder.

**Impact of Scaling.** We observe that continuous representations, semantic tokenizers, and, to some extent, hybrid tokenizers maintain stable performance across different model architectures and data scales, whereas acoustic tokenizers are much more sensitive to these factors. We organize our findings into two parts: the effect of data scaling, and model scaling.

*Scaling Data.* Performance consistently improves with larger datasets, particularly for acoustic tokenizers. For example, Discrete WavLM achieves 6.0% WER on LibriSpeech (960h), 22.0% on Basque (116h), and 58.9% on Welsh (8h) at low bitrate with a BiLSTM head, illustrating a clear correlation between training data size and WER. Acoustic models such as EnCodec and DAC are particularly sensitive in low-resource settings, while continuous representations maintain strong performance even with limited data. Similar trends are observed in tasks like music genre classification and sound event classification, where discrete tokenizers degrade more sharply as training data shrinks.

*Scaling Model Architecture.* Increasing model capacity stabilizes training and improves performance, especially for acoustic tokenizers. In ASR tasks, switching from a BiLSTM to a Branchformer significantly reduces WER, with larger relative gains observed for acoustic models (e.g., EnCodec achieves 8% WER with Branchformer versus 16% with BiLSTM). Similarly, in intent classification, moving from a shallow linear+statistical pooling model to a deeper BiLSTM, and in TTS, replacing Tokotron with VALL-E, both lead to improvements, particularly for acoustic tokenizers that otherwise struggle with shallow architectures. When training data is limited, deeper models often fail to converge with discrete tokens, particularly for acoustic tokenizers, making it difficult to benefit from increased model capacity.

**Hyperparameter Sensitivity and Stability.** High-bitrate acoustic models exhibit greater training instability, with larger variance across random seeds, particularly when using shallow architectures. This instability likely stems from the optimization challenges associated with managing multiple codebooks and the informational redundancy in high-bitrate codes, which small datasets cannot effectively leverage. Deeper models, such as Branchformer for ASR and VALL-E for TTS, substantially improve stability, although large-vocabulary models like SQ-Codec (around 20k tokens) can still struggle to converge due to increased prediction complexity.

Acoustic tokenizers are highly sensitive to data scale and model capacity, often failing to converge in low-resource settings. Semantic tokenizers offer better stability with limited data but still lag behind continuous representations when data is extremely scarce. *Careful tuning and appropriate scaling of both data and model architecture are crucial when using acoustic tokens, especially in low-resource or high-bitrate scenarios.*

## 5.3 Ranking

We present rankings that summarize the overall performance of audio tokenizers across high-level task categories. To compute these rankings, tokenizers are first ordered by their performance on each metric within a task (with rank N as best and rank 1 as worst for N tokenizers), and the ranks are then averaged across all relevant tasks in each category. For tasks with multiple metrics, rankings are computed per metric and averaged to obtain the final score. Speech downstream tasks are divided into Discriminative and Generative. Discriminative tasks are further split into Content-level, which require higher-level semantic understanding (e.g., ASR, intent classification, keyword spotting), and Acoustic-level, which rely on fine-grained acoustic cues (e.g., emotion recognition, speaker identification, speaker verification).

The figure 3 highlights overall trends and the relative strengths and weaknesses of different tokenizers across domains. No tokenizer consistently dominates all categories, performance varies by task and domain. In

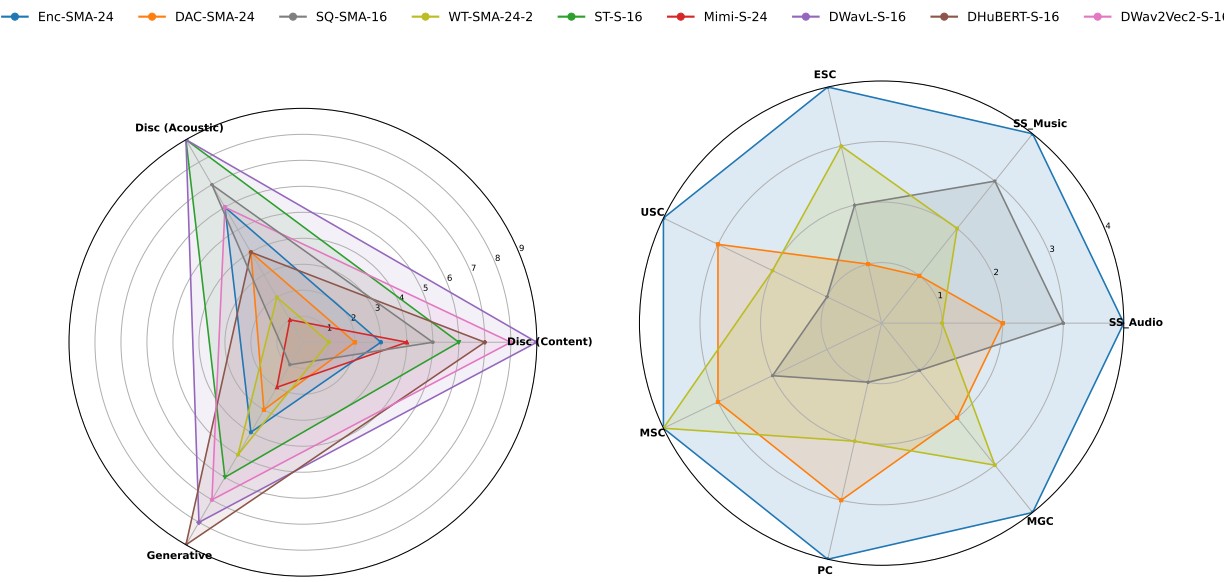

Figure 3: Radar plots of average rankings of audio tokenizers across task categories. Higher values indicate better overall performance. The right plot shows results for speech tasks, where Discrete WavLM achieves the strongest performance. The left plot shows results for music and audio tasks, where EnCodec performs best.

speech, Discrete WavLM achieves the highest overall ranking, while in music and general audio, EnCodec performs best. These rankings are intended as a high-level overview of performance trends rather than strict recommendations. For practical applications, we encourage consulting the full benchmark tables and task-specific analyses to select the most suitable tokenizer. A key factor lies in the type of information preserved during quantization. For example, in tasks such as ASR, where linguistic content is critical, we observe that semantic tokenizers derived from self-supervised models (e.g., HuBERT, WavLM, wav2vec 2.0) achieve the best performance, followed by approaches that incorporate SSL distillation into the codebook (e.g., SpeechTokenizer, Mimi). In contrast, for tasks such as speaker identification, where fine-grained acoustic cues are essential, tokenizers designed to retain detailed signal information for reconstruction (e.g., EnCodec, DAC) perform better. For general audio and music tasks, EnCodec consistently outperforms other tokenizers across bitrates and domains, while DAC tends to underperform, likely due to its emphasis on perceptual quality over signal fidelity. We also note that additional factors such as sampling rate, training data, and model scale can influence performance. These aspects have been studied in prior work (Mousavi et al., 2025) and are beyond the scope of our study, as we focus on comparing off-the-shelf tokenizers under a unified evaluation framework.

## 5.4 Tokenizer Category Trade-offs

We provide a qualitative comparison of acoustic, semantic, and hybrid tokenizers across key evaluation dimensions, including reconstruction quality, downstream performance, and robustness in low-resource settings (Table 5). This comparison highlights the strengths and limitations of each category, offering guidance for selecting appropriate tokenizers based on task requirements and resource constraints.

Table 5: Qualitative comparison of tokenizer categories across key evaluation dimensions.

| Tokenizer Type | Reconstruction | Downstream | Low-Resource Robustness | Model Scalability | Convergence Speed | efficiency |
|---|---|---|---|---|---|---|
| Acoustic | High | Low | ✗ | Requires large models | Slow | High |
| Semantic | Low | High | ✓ | Can work with smaller models | Fast | Low |
| Hybrid | Moderate | Moderate | ✗ | Depends on data | Moderate | Moderate |

## 6 Conclusion

We introduced DASB, a benchmark for evaluating discrete audio tokens across a range of tasks and domains. Our results show that semantic tokens generally outperform acoustic tokens in both generative and discriminative tasks. Scaling data and model capacity helps discrete representations narrow the gap with continuous self-supervised features. However, in low-resource settings, scaling is not feasible, and discrete tokens, especially acoustic ones, struggle to match the robustness of continuous representations. This highlights the need for future work on improving tokenization methods to better preserve phonetic, semantic, and paralinguistic information for integration into multimodal language models.

## 7 Acknowledgements

We acknowledge the support of the Natural Sciences and Engineering Research Council of Canada (NSERC) and the Digital Research Alliance of Canada (alliancecan.ca). We thank NVIDIA for donating part of the GPUs needed for this work through the NVIDIA Academic Grant Program.

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

# A  General Information

## A.1  Computational Resources

We designed our benchmark to be computationally accessible. Every task runs on GPUs with 32 GB or more of VRAM. Tasks like keyword spotting takes only 8 hours, while TTS might require about 48 hours on a single NVIDIA A100 GPU.

## A.2  Impact

We believe DASB can have a positive impact on the research community. We do not foresee a direct negative societal impact or misuse of our benchmark. However, we acknowledge that DASB can potentially accelerate progress in multi-modal large language models, which, in turn, have a wide range of potential positive and negative uses that society is still working to assess.

## A.3  Hosting and Maintenance Plan

DASB is a community-driven and open-source initiative. We plan to extend it by running additional experiments and including new audio tokenizers and tasks. We welcome external contributors.

## A.4  Licensing

Our work is licensed under Apache 2.0 (`https://www.apache.org/licenses/LICENSE-2.0`).

Table 6: Licenses for the models used in our benchmark.

| Model | License |
|---|---|
| HuBERT-large | Apache 2.0 |
| WavLM-large | ATTRIBUTION-SHAREALIKE 2.0 |
| Wav2vec2-large | Apache 2.0 |
| EnCodec | MIT license |
| DAC | MIT license |
| SpeechTokenizer | Apache 2.0 |
| SQCodec | MIT license |
| Mimi | CC-BY-4.0 |
| WavTokenizer | MIT license |

## A.5  Author Statement

We, the authors, will bear all responsibility in case of violation of rights.

# B  Discrete Audio Models Details

Additional information about the tokenizers used in our benchmark is provided in Table 7.

Table 7: Features of the Considered Discrete Audio Encoders.

| Model | Dataset | Repo |
|---|---|---|
| Discrete Hubert (Mousavi et al., 2024) | LibriSpeech960(Korvas et al., 2014) | huggingface.co/poonehmousavi/SSL_Quantization |
| Discrete WavLM (Mousavi et al., 2024) | LibriSpeech960(Korvas et al., 2014) | huggingface.co/poonehmousavi/SSL_Quantization |
| Discrete Wav2Vec2 (Mousavi et al., 2024) | LibriSpeech960(Korvas et al., 2014) | huggingface.co/poonehmousavi/SSL_Quantization |
| EnCodec (Défossez et al., 2023) | DNS (Dubey et al., 2024), CommonVoice (Ardila et al., 2020), AudioSet (Gemmeke et al., 2017), FSD50K (Fonseca et al., 2021), and Jamendo (Bogdanov et al., 2019) | github.com/facebookresearch/encodec |
| DAC (Kumar et al., 2023) | DAPS(Mysore, 2014), DNS (Dubey et al., 2024), CommonVoice (Ardila et al., 2020), VCTK (Yamagishi et al., 2019), MUSDB (Rafii et al., 2017b), and Jamendo (Bogdanov et al., 2019) | github.com/descriptinc/descript-audio-codec |
| SpeechTokenizer (Zhang et al., 2024) | LibriSpeech960(Korvas et al., 2014) | github.com/ZhangXInFD/SpeechTokenizer |
| Mimi (Défossez et al., 2024) | Unsupervised English speech | huggingface.co/kyutai/mimi |
| SQCodec (Yang et al., 2024a) | Multilingual LibriSpeech (MLS)  (Korvas et al., 2014) | huggingface.co/Dongchao/UniAudio |
| WavTokenizer (Ji et al., 2024) | LibriTTS(Zen et al., 2019), CommonVoice (Ardila et al., 2020), VCTK (Yamagishi et al., 2019), AudioSet (Gemmeke et al., 2017), and Jamendo (Bogdanov et al., 2019) | huggingface.co/novateur/WavTokenizer-medium-music-audio-75token |

# C  Dataset and Downstream Models

Table 8 provides a summary of the datasets and the two downstream architectures used for each task.

Table 8: Dataset and Downstream Models

| Dataset | Task | 1st Architecture | 2nd Architecture | Dataset Link |
|---|---|---|---|---|
| LibriSpeech (Korvas et al., 2014) | Speech Recognition | BiLSTM | Bracnhformer | openslr.org/12 |
| CommonVoice 17.0 (Ardila et al., 2020) | Speech Recognition | BiLSTM | Bracnhformer | commonvoice.mozilla.org/en/datasets |
| VoxCeleb1 (Nagrani et al., 2017) | speaker verification/identification | ECAPA-TDNN | BiLSTM + Linear | robots.ox.ac.uk/ vgg/data/voxceleb/vox1.html |
| IEMOCAP (Busso et al., 2008) | Emotion Recognition | ECAPA-TDNN | Time-Pooling + Linear | sail.usc.edu/iemocap/ |
| Speech Commands (Warden, 2018) | Keyword Spotting | ECAPA-TDNN | X -Vectors | tensorflow.org/datasets/catalog/speech_commands |
| SLURP (Bastianelli et al., 2020) | Intent Classification | BiLSTM + Linear | Time-Pooling + Linear | zenodo.org/record/4274930 |
| VoiceBank (Valentini-Botinhao et al., 2016) | Speech Enhancement | Conformer | CRDNN | datashare.ed.ac.uk/handle/10283/2791 |
| Libri2Mix (Cosentino et al., 2020) | Speech Separation | Conformer | CRDNN | github.com/JorisCos/LibriMix |
| LibriTTS(Zen et al., 2019)/LJSpeech (Ito, 2017) | Text-to-Speech | VALL-E | Shallow Transformer | LibriTTS: : openslr.org/60// / LJ:Speech: keithito.com/LJ-Speech-Dataset/ |
| FUSS (Kavalerov et al., 2019) | Audio Source Separation | Conformer | CRDNN | github.com/google-research/sound-separation/blob/master/datasets/fuss/ |
| MUSDB (Rafii et al., 2017a) | Music Source Separation | Conformer | CRDNN | sigsep.github.io/datasets/musdb.html |
| ESC50 (Piczak, 2015) | Sound Classification | ECAPA-TDNN | Linear | https://github.com/karolpiczak/ESC-50 |
| GTZAN (Tzanetakis & Cook, 2002) | Music Genre Classification | ECAPA-TDNN | Linear | huggingface.co/datasets/marsyas/gtzan |
| UrbanSound8K (Ooi et al., 2021) | Event Sound Classification | ECAPA-TDNN | Xvector | https://github.com/ravising-h/Urbansound8k |
| GTZAN Music-Speech (Tzanetakis, 1999) | Music vs Speech Classification | ECAPA-TDNN | Xvector | https://github.com/tensorflow/datasets/blob/master/docs/catalog/gtzan$_{music_speech.md}$ |
| NSynth (Engel et al., 2017) | Pitch Classification | ECAPA-TDNN | Xvector | https://magenta.tensorflow.org/datasets/nsynth |

# D  Second Probing Head Result

Tables 9, 10 and 11 show the results obtained with second downstream architectures. Note that table 8 indicates the first and second architectures explored for each task.

For English ASR, the Branchformer consistently outperforms the BiLSTM across most tokenizers, with the exception of SQ-Codec. This deviation is likely due to two main factors: first, the use of cross-entropy loss in Branchformer versus CTC loss in BiLSTM; and second, the exceptionally large vocabulary size of SQ-Codec (~20k tokens), which makes optimization more challenging under cross-entropy. Additionally, SQ-Codec was originally developed for diffusion-based models rather than standard autoregressive transformer architectures, which may further limit its compatibility with sequence-to-sequence training objectives. We observe a similar trend for SQ-Codec in other tasks as well, including SE and TTS.

The second TTS architecture we use is a simple autoregressive encoder-decoder transformer with a single linear prediction head that jointly predicts all codebooks. While this setup is effective for single-layer continuous features and tokenizers that retain high-level representations, it does not scale well to low-level representations that require many codebooks for high-fidelity reconstruction (e.g., acoustic tokenizers). This has led to non-convergence in some configurations. Architectures with built-in residual prediction, such as VALL-E, are better suited for handling such complex tokenizations.

Table 9: DASB results for discriminative tasks (speech) with the second downstream architecture.

| Models\Tasks | #Q | ASR-En WER↓ | | ASR-LR WER↓ | | ER ACC↑ | IC ACC↑ | KS ACC↑ | SI ACC↑ | SV EER↓ |
|---|---|---|---|---|---|---|---|---|---|---|
| | | Clean | Other | Welsh | Basque | | | | | |
| Continuous | – | 4.07 | 6.81 | 34.98 | 13.51 | 68.60 | 75.20 | 98.70 | 99.02 | 5.817447 |
| Enc-SMA-24 | 2 | 12.70±0.37 | 29.09±0.13 | 100.00±1.41 | 41.56±0.65 | 42.10±1.60 | 19.47±0.05 | 68.53±5.22 | 84.42±12.45 | 21.11±0.21 |
| | 8 | 8.43±0.13 | 21.77± 0.36 | 99.33±0.30 | 56.28±26.46 | 44.73±1.56 | 18.30±0.36 | 68.53±1.92 | 78.80±4.49 | 15.76±0.09 |
| | 32 | 9.95±1.17 | 23.24± 1.22 | 99.85±0.21 | 63.13±0.91 | 43.47±1.27 | 17.30±0.22 | 57.23±2.35 | 80.12±7.98 | 20.27±1.78 |
| DAC-SMA-24 | 2 | 14.84±0.25 | 33.88±0.20 | 98.04±0.26 | 48.37±4.97 | 45.20±0.54 | 17.10±0.42 | 68.80±6.30 | 78.98±10.61 | 23.00±0.04 |
| | 8 | 10.73± 0.10 | 25.39± 0.20 | 97.29±0.56 | 44.53±3.22 | 44.00±3.42 | 16.50±0.10 | 61.83±2.85 | 87.62±3.23 | 16.38±0.68 |
| | 32 | 13.13±0.16 | 28.47±0.19 | 98.04±0.57 | 47.17± 40.434578 | 35.23±3.91 | 15.30±0.10 | 57.23±2.35 | 73.00±2.50 | 21.04±0.45 |
| ST-S-16 | 2 | 9.48±0.10 | 22.68±0.10 | 99.31±0.06 | 36.95±0.31 | 56.13±1.19 | 49.00±0.14 | 87.30±3.08 | 77.20±7.51 | 21.18±0.43 |
| | 8 | 9.06± 0.45 | 21.72±0.23 | 99.83±0.84 | 33.28±0.93 | 56.37±1.63 | 47.07±0.26 | 92.69±0.41 | 81.20±0.22 | 19.24±0.44 |
| Mimi-S-24 | 8 | 9.73±0.61 | 22.65±0.41 | 96.60±0.46 | 39.32±1.38 | 46.90±1.22 | 44.80±0.37 | 92.15±0.67 | 76.03±0.17 | 17.27±0.08 |
| | 32 | 10.84±0.56 | 24.10±0.36 | 97.47±0.67 | 43.90±2.45 | 42.73±2.67 | 35.90±0.45 | 90.04±0.60 | 62.33±0.96 | 21.15±1.09 |
| DWavL-S-16 | 2 | 4.78±0.25 | 10.58±0.17 | 84.18±4.98 | 22.53±0.34 | 60.67±0.70 | 64.67±0.05 | 96.79±0.79 | 88.71±4.96 | 21.29±0.72 |
| | 6 | 5.07±0.17 | 9.57±0.20 | 84.01±2.60 | **21.69±1.34** | **61.20±1.60** | **70.27±0.42** | **97.21±0.43** | 92.13±4.78 | 12.95±0.37 |
| DHuBERT-S-16 | 2 | 4.63±0.15 | 10.38±0.09 | **74.46±1.30** | 23.36±0.57 | 58.53±0.60 | 57.23±0.17 | 63.53±2.18 | 82.50±8.20 | 17.84±0.68 |
| | 6 | **4.02±0.31** | **8.96± 0.23** | 82.62±1.34 | 32.87±15.07 | 59.40±1.70 | 61.90±0.08 | 83.93±1.35 | 89.06±4.04 | 12.00±0.13 |
| DWav2Vec2-S-16 | 2 | 8.44±0.17 | 19.57±0.22 | 88.60±3.46 | 29.05±0.64 | 59.10±1.99 | 51.93±0.12 | 92.28±0.22 | 74.53±0.090 | 21.04±0.52 |
| | 6 | 5.52± 0.12 | 12.52±0.15 | 92.86±1.88 | 25.37± 0.20 | 59.73±0.83 | 61.10±0.36 | 95.41±1.02 | 80.65±0.07 | **11.63±0.19** |
| SQ-SMA-16 | 4 | 91.57±0.49 | 92.90±0.41 | 97.68±0.30 | 98.92±0.17 | 40.93±1.18 | 16.27±0.05 | 85.47±3.24 | 93.24±1.21 | 13.93±0.57 |
| WT-SMA-24-2 | 1 | 16.11±0.18 | 35.48±0.35 | 99.45±0.04 | 52.63±0.76 | 55.17±0.09 | 16.23±0.09 | 71.27±1.64 | **94.53±6.25** | 17.58±0.48 |

Table 10: DASB results for generative tasks (speech) with the second downstream architecture.

| Models\Tasks | #Q | SE | | | SS - Speech | | | | TTS | |
|---|---|---|---|---|---|---|---|---|---|---|
| | | DNSMOS ↑ | dWER ↓ | Spk Sim↑ | DNSMOS Rec↑ | DNSMOS Sep↑ | dWER ↓ | Spk Sim↑ | UTMOS ↑ | dWER ↓ |
| Continuous | – | **3.49** | **4.92** | **0.93** | – | **3.43** | **24.42** | **0.873** | **4.08** | **4.76** |
| Enc-SMA-24 | 2 | 3.17 ± 0.01 | 36.78 ± 1.15 | 0.85 ± 0.00 | 3.19 | 3.11±0.01 | 86.95±0.74 | 0.87±0.00 | 1.34 ± 0.00 | 53.93±0.99 |
| | 8 | 2.94 ± 0.01 | 50.32 ± 2.81 | 0.84 ± 0.01 | 3.54 | 3.05±0.00 | 67.65±2.21 | 0.88±0.00 | – | – |
| | 32 | 3.00 ± 0.02 | 108.65 ± 9.64 | 0.76 ± 0.01 | 3.72 | 3.00±0.00 | 98.82±2.65 | 0.83±0.00 | – | – |
| DAC-SMA-24 | 2 | 3.31 ± 0.01 | 61.72 ± 2.54 | 0.86 ± 0.00 | 3.16 | 2.97±0.01 | 111.5±0.50 | 0.839±0.00 | 1.79±0.03 | 36.93±6.05 |
| | 8 | 2.91 ± 0.10 | 131.33 ± 2.08 | 0.68 ± 0.03 | 3.67 | 3.16±0.01 | 82.33±1.42 | 0.892±0.00 | 1.61±0.05 | 102.08±2.95 |
| | 32 | 2.84 ± 0.02 | 103.88 ± 8.79 | 0.78 ± 0.01 | 3.76 | 2.66±0.09 | 178.0±25.0 | 0.80±0.01 | – | – |
| ST-S-16 | 2 | 3.15 ± 0.01 | 31.76 ± 0.74 | 0.85 ± 0.00 | 3.20 | 3.12±0.01 | 89.9±0.94 | 0.862±0.00 | 2.48±0.01 | 46.59±4.32 |
| | 8 | 3.30 ± 0.01 | 33.95 ± 3.29 | 0.85 ± 0.00 | 3.72 | 3.29±0.00 | 70.43±0.75 | 0.87±0.01 | 1.76±0.02 | 89.06±3.16 |
| Mimi-S-24 | 8 | 3.31 ± 0.01 | 107.27 ± 14.68 | 0.81 ± 0.01 | 3.37 | 3.39±0.00 | 109.00±2.16 | 0.806±0.00 | 1.89±0.03 | 25.26 ± 4.40 |
| | 32 | 2.93 ± 0.14 | 159.0 ± 35.59 | 0.75 ± 0.01 | 3.65 | 3.20±0.00 | 115.66±1.25 | 0.84±0.00 | 1.24±0.01 | 112.22±8.75 |
| DWavL-S-16 | 2 | 3.57 ± 0.01 | 27.64 ± 4.13 | 0.88 ± 0.00 | 3.57 | 2.97±0.01 | 50.36±1.05 | 0.839±0.00 | 3.01±0.02 | 4.99±4.61 |
| | 6 | **3.58 ± 0.00** | **9.39 ± 0.28** | **0.89 ± 0.00** | 3.75 | **3.77±0.00** | **33.51±0.53** | **0.896±0.00** | **3.87±0.02** | 13.35±2.20 |
| DHuBERT-S-16 | 2 | 3.46 ± 0.03 | 59.97 ± 3.33 | 0.85 ± 0.00 | 3.68 | 3.65±0.00 | 83.98±0.18 | 0.826±0.00 | 3.41±0.01 | **1.79±1.79** |
| | 6 | 3.50 ± 0.01 | 13.31 ± 0.09 | 0.88 ± 0.00 | 3.72 | 3.69±0.00 | 75.99±1.26 | .88±0.00 | 3.80±0.00 | 5.76±0.49 |
| DWav2Vec2-S-16 | 2 | 3.55 ± 0.01 | 30.91 ± 0.32 | 0.88 ± 0.00 | 3.61 | 3.64±0.01 | 94.88±1.98 | 0.805±0.00 | 2.26±0.02 | 26.26±1.26 |
| | 6 | 3.57 ± 0.01 | 13.06 ± 0.40 | **0.89 ± 0.00** | 3.77 | 3.74±0.01 | 74.59±1.40 | 0.88±0.00 | 3.07±0.01 | 13.96±0.32 |
| SQ-SMA-16 | 4 | 3.09 ± 0.04 | 328.0 ± 35.00 | 0.76 ± 0.01 | **3.77** | 3.59±0.00 | 72.39±0.6 | **0.90±0.00** | – | – |
| WT-SMA-24 | 1 | 3.36 ± 0.01 | 62.15 ± 2.52 | 0.85 ± 0.00 | 3.57 | 3.43±0.01 | 128.33±2.49 | 0.85±0.00 | 2.44±0.01 | 14.09±0.84 |
| Mixture | – | | | | – | 3.43 | – | – | | |

Table 11: DASB results for generative and discriminative tasks (music - audio) with the second downstream architecture.

| Models\Tasks | #Q | SS - Audio | | SS - Music | | | | ESC | USC | MSC | PC | MGC |
|---|---|---|---|---|---|---|---|---|---|---|---|---|
| | | SI-SDRi↑ Rec | Sep | SI-SDRi↑ Rec | Sep | SAR↑ | SIR↑ | ACC↑ | ACC↑ | ACC↑ | ACC↑ | ACC↑ |
| Continuous | – | – | **15.07** | – | **13.29** | **9.56** | **11.99** | **94.50** | **87.60** | **100.00** | **90.80** | **87.00** |
| Enc-SMA-24 | 2 | 0.76 | **5.57±0.49** | 3.36 | **-0.03 ± 1.46** | **-3.78 ± 1.19** | **4.76 ± 1.13** | 34.83±0.47 | 25.80±4.47 | 97.22±3.93 | 71.40±0.59 | **70.33±1.70** |
| | 8 | 3.87 | 0.04 ± 0.07 | 7.99 | -11.96 ± 0.38 | -15.55 ± 0.32 | -0.16 ± 0.1 | **37.00±0.73** | 30.30±4.93 | **100.00±0.00** | **72.13±0.74** | 54.67±3.86 |
| | 32 | **5.76** | -3.25 ± 0.11 | **11.10** | -11.27 ± 0.09 | -15.75 ± 0.08 | -0.35 ± 0.13 | 35.43±1.45 | 24.07±3.60 | 97.22±3.93 | 67.27±1.46 | 39.67±1.25 |
| DAC-SMA-24 | 2 | 0.12 | 2.58 ± 0.02 | 2.37 | -4.98 ± 0.08 | -9.31 ± 0.1 | 3.55 ± 0.03 | 31.03±1.84 | **60.03±2.24** | **100.00±0.00** | 56.73±0.38 | 50.00±0.82 |
| | 8 | 3.33 | -0.06 ± 0.53 | 6.66 | -8.19 ± 0.76 | -11.61 ± 0.75 | 0.84 ± 0.59 | 28.60±0.79 | 21.50±2.25 | 91.67±0.00 | 59.40±0.59 | 47.67±3.09 |
| | 32 | 4.73 | -7.8 ± 0.15 | 8.54 | -11.55 ± 0.18 | -11.76 ± 0.53 | -3.29 ± 0.86 | 25.77±0.82 | 15.07±1.35 | 100.00±0.00 | 48.87±0.90 | 39.67±1.25 |
| SQ-SMA-16 | 4 | 3.62 | 5.18 ± 0.25 | 5.53 | -1.78 ± 0.08 | -4.71 ± 0.19 | 3.77 ± 0.04 | 31.37±1.370 | 13.20±0.86 | 91.67±0.00 | 49.20±2.16 | 42.67±0.47 |
| WT-SMA-24-2 | 1 | -24.05 | -15.49 ± 0.87 | -2.66 | -3.68 ± 1.03 | -7.49 ± 0.88 | 3.03 ± 0.79 | 34.50±0.82 | 18.03±2.76 | 97.22±3.93 | 48.87±1.79 | 48.00±1.41 |
| Mixture | – | – | -16.5 | – | -7.71 | 50.01 | -inf | – | – | – | – | – |

# E  Effect of Embedding initialization

We study different configurations for initializing the embedding layers of audio tokens (Table 12). Two options are considered: (1) random initialization of the embedding layers, and (2) initialization using the encoder's embedding layer, without freezing it during training. We evaluate both setups on two tasks: ASR LibriSpeech (In-domain, as the dataset is included in all the tokenizer's training data) and keyword spotting on the Google Speech Commands dataset (out-of-distribution). With regard to in-domain vs. out-of-domain performance, we hypothesized that encoder-based initialization would be more beneficial in the in-domain setting. Our observations partially support this: semantic and hybrid tokenizers appear more sensitive to domain shift, and initialization can sometimes harm generalization on out-of-distribution tasks. In contrast, acoustic tokenizers tend to benefit more consistently from initialization. Consistent with findings in (Mousavi et al., 2024), semantic tokenizers tend to perform better with random initialization, while acoustic tokenizers improve with encoder-based initialization. However, this trend may also reflect architectural differences. For example, encoder embeddings are often lower-dimensional (e.g., 128), requiring an additional projection layer to match the 1024-dimensional downstream embeddings. This added capacity may itself contribute to improved performance, independent of initialization strategy. To ensure consistency and fairness, we adopt random initialization for all embeddings when reporting final results. In the case of SQ-Codec, the learnable embedding configuration failed to converge, likely due to its large vocabulary size (~20k tokens). As a

workaround, we use a ternary matrix-based dequantization scheme to represent its token embeddings. Finally, to eliminate optimization bias, we perform separate learning rate tuning for all embedding initialization experiments.

Table 12: Comparison of random initialization vs. encoder-based initialization (pretrained + fine-tuned) on ASR (test-clean/test-other, BiLSTM) and keyword spotting (ECAPATDNN).

| Models\Tasks | #Q | ASR (WER) | | KS (ACC) | |
|---|---|---|---|---|---|
| | | Random | Initialized | Random | Initialized |
| Enc-SMA-24 | 8 | 16.83 / 38.98 | 10.69 / 28.64 | 68.53 | 76.50 |
| DAC-SMA-24 | 8 | 16.29 / 39.57 | 15.52 / 37.54 | 61.83 | 77.60 |
| ST-S-16 | 8 | 12.69 /30.99 | 8.56 / 24.44 | 92.69 | 87.60 |
| Mimi-S-24 | 8 | 13.96 / 31.53 | 11.86 / 28.18 | 92.14 | 91.17 |
| DWavL-S-16 | 6 | 4.32 / 10.73 | 3.77 / 9.79 | 97.20 | 87.50 |
| WT-SMA-24 | 1 | 28.69 / 52.90 | 20.7 / 43.99 | 71.26 | 75.50 |

## F   Reconstruction Analysis

We use the CodecSUPERB benchmark and VRESA results to evaluate reconstruction quality. Table 13 summarizes the result on the Librispeech test-clean set. UTMOS is used as the primary perceptual metric. For application-level evaluation, the resynthesized audio is passed through pre-trained models for ASR and speaker identification (SPK), with WER and accuracy as the evaluation metrics respectively. Overall, DAC and SQ-Codec achieve the highest average performance. Interestingly, these same models rank among the lowest in our downstream evaluations and show high variability. This contrast is particularly noteworthy. While semantic tokenizers typically outperform hybrid and acoustic ones in downstream tasks, the trend is reversed in reconstruction-based evaluations. Additionally, we observe that increasing the number of codebooks improves reconstruction quality. In contrast, in downstream tasks, more codebooks often reduce performance due to added redundancy and complexity. These findings suggest that strong reconstruction quality does not necessarily indicate better preservation of semantic or task-relevant information. One reason for this is the role of the decoder. Since most reconstruction losses are applied at the decoder output during training, the decoder plays a significant role in shaping the perceptual quality of the resynthesized audio. As a result, reconstruction performance may reflect decoder strength more than the quality of the token representations. In general, reconstruction metrics remain essential for evaluating tokenizers in generative settings or transmission scenarios, where the output is audio and fidelity and intelligibility are critical. These findings highlight the importance of evaluating discrete tokens from multiple perspectives. While discrete tokenizers offer clear advantages in modularity and efficiency, there is still a performance gap compared to continuous features. Bridging this gap is crucial for establishing discrete tokens as a viable alternative in future multimodal language models.

## G   Hyperparameters

For English ASR, we compare character-level and 500-unit byte pair encoding (BPE) tokenizations, selecting the better-performing option. For multilingual datasets, BPE segmentations with 100, 200, or 500 units are chosen based on dataset size and tokenizer characteristics. The best-performing setting is reported.

*General TTS Scaling Observations.* Similar to the approach used in ESPNet's (Watanabe et al., 2018) implementation, during inference, we generate 10 independent samples using top-K sampling and then, using Whisper Large (Radford et al., 2022), select the one that best matches the text label. While this technique significantly reduces variability and brings most tokenizers into a dWER range comparable to that of mainstream generally available systems, notably, it does not sufficiently improve performance for high-bandwidth configuration, indicating a fundamental difficulty with predicting a large number of codebooks in each step rather than a simple issue of high sampling variability. Without multi-sampling, we observe that a much wider gap between semantic and acoustic tokens in word error rates.

Table 13: CodecSUPERB benchmark performance on the LibriSpeech test-clean set.

| Models | #Q | UTMOS ↑ | WER (beam=5) ↓ | Spk Sim↑ |
|---|---|---|---|---|
| Ground truth | - | 4.09 | 2.83 | 100.00 |
| Enc-SMA-24 | 2 | 1.58 | 5.44 | 41.70 |
|  | 8 | 3.09 | 2.78 | 71.74 |
|  | 32 | 3.74 | 2.77 | 78.19 |
| DAC-SMA-24 | 2 | 1.67 | 9.59 | 44.94 |
|  | 8 | 3.83 | 2.41 | 85.65 |
|  | 32 | **4.05** | 2.72 | 79.89 |
| ST-S-16 | 2 | 2.32 | 4.20 | 34.76 |
|  | 8 | 3.60 | 3.53 | 72.66 |
| Mimi-S-24 | 8 | 3.60 | 3.72 | 69.72 |
|  | 32 | 3.91 | 2.96 | 85.1 |
| DWavL-S-16 | 2 | 3.31 | 4.97 | 32.55 |
|  | 6 | 3.31 | 4.34 | 34.86 |
| DHuBERT-S-16 | 2 | 3.43 | 7.18 | 28.64 |
|  | 6 | 3.71 | 5.52 | 31.35 |
| DWav2Vec2-S-16 | 2 | 2.75 | 14.12 | 18.29 |
|  | 6 | 3.69 | 3.99 | 31.79 |
| SQ-SMA-16 | 4 | 3.90 | **2.37** | **86.66** |
| WT-SMA-24-2 | 1 | 3.77 | 8.10 | 59.89 |

## H   TTS Inference Hyperparameter Search

The TTS task, in particular, proved rather sensitive to the selection of certain hyperparameters in preliminary experiments; furthermore, given VALL-E is a generative model conditioned on text tokens - with top-K sampling used during inference, the selection of hyperparameters involves a trade-off between natrualness and fidelity to the text conditioning.

We perform a grid search of combinations of the the $k$ hyperparameter in $top-K$ and the sampling temperature using a random sample of 100 speech samples.

Figure 4: TTS Inference Hyperparameter Search - UTMOS

Figuees 4 and 5 show the relationship between k, sampling temperatures and the UTMOS and dWER observed (simplified sampling, no post-selection). We observe empirically that for most tokenizers, there appears to be an optimal setting for both hyperparameters, and performance can be affected significantly by the selection. The selection of optimal parameters is not universal and, rather, is, tokenizer-dependent. The effect of the sampling temperature and the k chosen on UTMOS appears to be rather minor for most tokenizers except at the extremes, whereas the effect on dWER can be very significant, which is consistent

Figure 5: TTS Inference Hyperparameter Search - dWER

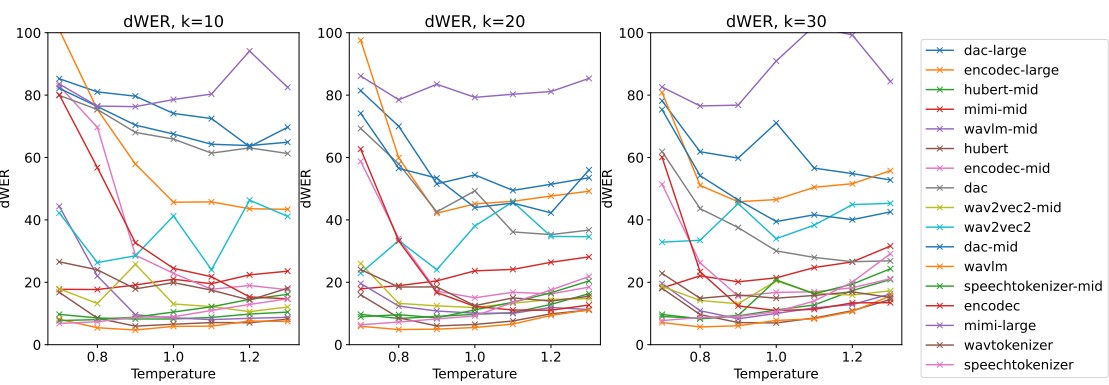

with the intuitive expectation that the parameters control primarily whether the extent to which the tokens from tail ends of the distribution will be explored. The lower the k and the temperature, the closer the sample is to selecting only the most likely tokens given the conditioning, the higher the parameters, the more diversity is allowed. UTMOS being a measure of vocal quality is more related to the audio quality of the tokenizer's reconstruction and will be affected minimally if the model mispronounces a word or emits a non-word sequence of sounds. We also observe that for many tokenizers the relationship between the temperature and the dWER is approximately convex, especially at the medium k selection (k = 20). One likely explanation for the observed behaviour is that insufficient sampling diversity may cause instability in autoregressive influence, i.e. once a prediction error is made, it is more difficult for recover; however, too much tail-end sampling may lead to sequences with a low overall likelihood given the conditioning (i.e. utterances loosely inspired by the text rather than reproducing it with fidelity).

