# OpenReview forum: "DASB - Discrete Audio and Speech Benchmark"
_TMLR — Accepted by TMLR_

### Review · Reviewer_Rjak · 2025-11-08

**Summary Of Contributions:**

The paper introduces DASB (Discrete Audio and Speech Benchmark) — a large-scale, comprehensive benchmark designed to systematically evaluate discrete audio tokenizers across speech, general audio, and music domains. Its goal is to provide a unified and reproducible framework for assessing the trade-offs between discrete and continuous audio representations across both discriminative (e.g., ASR, emotion recognition) and generative (e.g., TTS, separation, enhancement) tasks.

Key contributions include:
- Direct evaluation of discrete tokens: Unlike prior work that relies on decoding to waveforms, DASB isolates the information loss from tokenization itself, enabling a more faithful assessment of discrete representations.
- Comprehensive benchmark coverage: DASB spans multiple domains and tasks with standardized evaluation protocols, supporting semantic, acoustic, and hybrid tokenizers.
- Extensive ablations and hyperparameter tuning: The paper studies how bitrate, codebook size, and model architecture affect downstream performance, offering practical guidance for future tokenizer design.
- Quantitative comparison with continuous features: The benchmark establishes a clear performance gap between discrete and continuous representations, identifying where discrete tokens remain insufficient.
- Open and reproducible framework: The authors release code, documentation, and leaderboards, encouraging community participation and standardization.

**Strengths**
- The paper unifies a fragmented research area, covering a broad range of models and tasks with strong experimental rigor (multi-seed, multi-architecture, controlled settings).
- The decision to evaluate tokens without decoding is methodologically sound and clarifies how much representational power is lost through quantization.
- The paper is well-structured, clearly explaining benchmark design, tasks, and implementation. Public release of code and leaderboard further enhances impact.
- Empirical findings (e.g., semantic tokens’ superiority for discriminative tasks, acoustic tokens’ speaker retention) offer practical guidance for both researchers and industry practitioners.
- The benchmark addresses a pressing need for standardization in the development of multimodel audio-language models.

**Weaknesses**
- The study is primarily empirical; it does not provide deeper explanations for why certain tokenization strategies succeed or fail.
- While the benchmark itself is new, much of the underlying setup (datasets, architectures) builds on existing benchmarks (e.g., SUPERB, SpeechBench) limiting the novelty of the contribution.
- Although encoder/decoder runtime is discussed, the benchmark does not deeply analyze latency or compression trade-offs under deployment constraints.
- The work evaluates known models rather than proposing new architectures or training objectives to improve tokenization, but this is okay since the primary contribution is a benchmark.
- The paper highlights that discrete tokens lag behind continuous representations but does not offer concrete strategies to bridge the gap.

Overall, this paper provides a valuable, timely, and well-executed benchmark that fills a crucial gap in evaluating discrete audio representations. Its primary contribution is infrastructural and diagnostic rather than algorithmic, but it is likely to have broad community impact by standardizing comparisons and informing the design of future multimodal audio-language systems.

**Audience:**

Yes

**Audience Explanation:**

TMLR’s readership includes researchers working on multimodal learning, representation learning, and self-supervised models. The paper’s focus on discrete audio tokens directly connects to current efforts to unify speech and text modalities (e.g., in models like Gemini, Qwen, and AudioLM). Understanding how tokenization choices affect performance is crucial for building scalable multimodal LLMs — a topic of high current relevance.

The community increasingly values standardized, reproducible benchmarks (e.g., SUPERB, HELM, BIG-Bench). This work’s DASB benchmark provides a rigorous, domain-spanning evaluation framework for discrete audio representations — a clear contribution to methodological best practices, which is directly aligned with TMLR’s scope.

**Broader Impact Concerns:**

The work does not include a broader impact statement. Some reflection on potential bias (coverage of linguistic, gender and cultural diversity) in the data and discussion on potential misuse of multimodal generative models would be useful.

**Claims And Evidence:**

Yes

**Claims Explanation:**

The authors make claims about (a) the comparative performance of discrete versus continuous representations, (b) the differing strengths of semantic, acoustic, and hybrid tokenizers, and (c) the impact of design factors such as bitrate and codebook size. These claims are convincingly supported by extensive quantitative experiments across a wide variety of datasets (speech, audio, music) and both discriminative and generative tasks. Tables 2–4 and multiple ablation studies back up each assertion with well-controlled comparisons. The authors use consistent evaluation settings and multiple random seeds, reducing variance-driven artifacts — a significant strength compared to prior benchmarks.

The benchmark pipeline is clearly explained, including tokenizer configurations, downstream architectures, hyperparameter tuning procedures, and evaluation metrics. The design choices (e.g., using the same downstream architecture for continuous and discrete baselines) make the comparisons methodologically fair and isolate the effect of discretization. Figures and tables are generally clear, though dense; the summary of patterns in Section 5 effectively synthesizes the evidence to support the main claims. The authors are careful not to overstate findings — they frame conclusions in terms of observed tendencies and practical implications rather than universal superiority.

While the empirical evidence is broad, it is mostly descriptive rather than explanatory. The evidence could be strengthened by deeper causal or statistical analysis.

**Requested Changes:**

The following changes would strengthen the work.

If I understand correctly, DASB measures representational utility rather than end-to-end model quality, and that findings may not directly transfer to models trained jointly with audio decoders. The authors should state that if so.

Discuss more dataset coverage and potential biases. Clarify that handling non-English, low-resource, or noisy-domain data would need further extensions.

Writing issues:
- Incorrect bracketing of citations. Please use \citep{} where appropriate.
- For references, please cite peer-reviewed versions wherever possible. E.g. for “Unity: Two-pass direct speech-to-speech translation with discrete units” by Inaguma et al. please use the ACL 2023 citation (and similarly for other references to arXiv versions).
- Appendix numbering is off. Appendix A is empty.

---

> ### Author Response · Authors · 2026-04-11
> **Explanatory Analysis**
>
> While our study is primarily empirical, we do provide insights into why different tokenization strategies succeed or fail, as discussed in Sections 5.1 and 5.4. A key factor lies in the type of information preserved during quantization. For example, in tasks such as ASR, where linguistic content is critical, we observe that semantic tokenizers derived from self-supervised models (e.g., HuBERT, WavLM, wav2vec 2.0) achieve the best performance, followed by approaches that incorporate SSL distillation into the codebook (e.g., SpeechTokenizer, Mimi). In contrast, for tasks such as speaker identification, where fine-grained acoustic cues are essential, tokenizers designed to retain detailed signal information for reconstruction (e.g., EnCodec, DAC) perform better. For general audio and music tasks, EnCodec consistently outperforms other tokenizers across bitrates and domains, while DAC tends to underperform, likely due to its emphasis on perceptual quality over signal fidelity. We also note that additional factors such as sampling rate, training data, and model scale can influence performance. These aspects have been studied in prior work[1] and are beyond the scope of our study, as we focus on comparing off-the-shelf tokenizers under a unified evaluation framework. We add clarification in section 5.3.
>
> [1] Mousavi, Pooneh, et al. "Discrete Audio Tokens: More Than a Survey!." arXiv preprint arXiv:2506.10274 (2025).

---

> > ### Author Response · Authors · 2026-04-11
> > **Novelty of the contribution**
> >
> > Discrete audio tokens can be evaluated from multiple perspectives[1] . As discussed in the related work section, several benchmarks have been proposed to assess discrete audio representations. For example, Codec-SUPERB evaluates reconstruction quality using subjective scores and downstream performance on synthesized audio, while ESPnet-Codec and VERSA provide toolkits for training and evaluating codec models. However, these benchmarks rely on decoding the discrete tokens back into audio before evaluation.
> >
> > In contrast, DASB is the first benchmark specifically designed to evaluate the direct use of discrete audio tokens on downstream tasks, without relying on waveform reconstruction. By removing the decoder, DASB isolates the information loss introduced by tokenization, allowing for a more accurate analysis of how discrete representations impact downstream tasks and multimodal language modeling.
> >
> > Main Differences Between DASB and Prior Benchmarks:
> >
> > - **Quantifies Information Loss from Tokenization**: By removing the decoder, DASB directly measures the information loss caused by quantization, enabling a clearer analysis of representation quality. While reconstruction-based evaluation is valuable, it can be affected by a strong decoder, especially since most tokenizer losses are applied to the decoder output, which may mask deficiencies in the token representations. As shown in Table 12 (Appendix F: Reconstruction Analysis), there is a clear trade-off between high-fidelity reconstruction and downstream task performance. Tokenizers optimized solely for reconstruction often fail to keep task-relevant features such as phonetic or semantic information. This limitation becomes particularly evident in tasks where the decoder is not involved (e.g., ASR or SLU). As discussed in [1] , different evaluation perspectives are complementary and necessary for a holistic understanding of discrete representations.
> >
> > - ** Insight into Lossy Nature of Discrete Representations for Multimodal Modeling as an alternative to continuous representations**: DASB allows for a systematic comparison between discrete and continuous representations. Our benchmark provides insight into how lossy quantized representations are, which is crucial for training multimodal models. The ultimate goal of such models is to have a unified representation capable of supporting various tasks. It is important that these representations are rich enough to be useful in this context. So far, we have observed that speech-aware multimodal models such as SALMON[2] and Qwen[3] prefer continuous features for tasks that require deep speech understanding, while generative models tend to prefer discrete tokens, as they are easier to train (converting regression tasks into classification). DASB helps clarify why certain models choose continuous over discrete representations by quantifying the trade-offs in information retention and task performance.
> >
> > - ** Broad Task and Tokenizer Coverage**: DASB covers both discriminative (ASR, intent, emotion, etc.) and generative (TTS, source separation, enhancement) tasks across speech, general audio, and music. Previous benchmarks typically focus on a narrower set of tasks or only one type of tokenizer (semantic or acoustic). In contrast, DASB evaluates a diverse range of tokenizers, including semantic, acoustic, and hybrid models.
> >
> > - **Practical Guidance and Task-Level Analysis**:DASB provides actionable insights for practitioners on which tokenizers are best suited for specific tasks and conditions. The analysis includes the effects of the number of codebooks, the downstream head, data scale, and initialization strategies.
> >
> > We emphasize that DASB is designed to complement, not replace, prior benchmarks, filling gaps in evaluation methodology and broadening our understanding of discrete tokenization. We have clarified this positioning and discussed relevant benchmarks in Section 2 (Related Work).

---

> > > ### Author Response · Authors · 2026-04-11
> > > **Latency Analysis**
> > >
> > > While a detailed deployment-focused analysis is beyond the scope of this work, we do provide several indicators related to latency and compression trade-offs. Specifically, Table 2 reports key properties such as frame rate and vocabulary size, which directly impact compression efficiency and processing cost. In addition, Figure 2 presents the time and memory required for encoding and decoding across tokenizers at different bitrates, providing an empirical comparison of their computational overhead. Moreover, our selection of tokenizers intentionally covers a range of latency–efficiency trade-offs, including higher-latency models (e.g., EnCodec, DAC) and lower-latency alternatives (e.g., WavTokenizer with a single codebook or tokenizers based on finite scalar quantization). We agree that a more comprehensive analysis under strict deployment constraints would be valuable, and we consider this an important direction for future work.

---

> > > ### Author Response · Authors · 2026-04-11
> > > **Directions for Closing the Gap Between Discrete and Continuous Representations**
> > >
> > > In this work, our primary goal is to systematically identify limitations and uncover general patterns rather than propose new methods. Our findings suggest that the performance gap largely depends on the type of information preserved during quantization. For example, in ASR tasks, where linguistic content is critical, semantic tokenizers derived from self-supervised models (e.g., HuBERT, WavLM, wav2vec 2.0) perform best, followed by approaches that incorporate SSL distillation into the codebook (e.g., SpeechTokenizer, Mimi). In contrast, for speaker identification, where fine-grained acoustic cues are essential, reconstruction-oriented tokenizers (e.g., EnCodec, DAC) achieve stronger performance. For general audio and music tasks, EnCodec consistently outperforms other tokenizers across bitrates and domains, while DAC tends to underperform, likely due to its emphasis on perceptual quality over signal fidelity. We believe these observations provide actionable insights and can guide future work toward designing tokenizers that better balance semantic and acoustic information, thereby helping to bridge the gap with continuous representations.

---

> > > > ### Author Response · Authors · 2026-04-11
> > > > **Explanatory Analysis**
> > > >
> > > > While our study is primarily empirical, we go beyond purely descriptive analysis by systematically quantifying key factors and linking them to observed performance trends. First, we provide consistent comparisons across different tokenizers for all tasks (Tables 2, 3, and 4). We further analyze the impact of architectural choices, such as the number of codebooks, and present a global ranking across tasks to highlight consistent patterns (Fig.\~3). These quantitative results are complemented by detailed discussions of observed trends in Sections 5.1 and 5.2, as well as a practical summary of key evaluation dimensions in Table 5. In addition, we include analyses of computational trade-offs (time and memory in Fig.~2) and ablation studies in the appendix (e.g., embedding initialization and the choice and scaling of downstream heads), which help isolate contributing factors.
> > > >
> > > > We would like to also emphasize that DASB enables a systematic comparison between discrete and continuous representations, providing insight into the extent of information loss introduced by quantization, an aspect that is crucial for training multimodal models. The ultimate goal of such models is to learn unified representations that can support a wide range of tasks. In this context, DASB offers actionable insights for practitioners by identifying where and why discrete representations lose information, and how this impacts downstream performance across diverse tasks and conditions. Our analysis further highlights the role of key factors such as the number of codebooks, downstream head design, data scale, and initialization strategies.

---

> > > > > ### Author Response · Authors · 2026-04-11
> > > > > **Requested Changes:**
> > > > >
> > > > > ### Comment 1
> > > > > > If I understand correctly, DASB measures representational utility rather than end-to-end model quality, and that findings may not directly transfer to models trained jointly with audio decoders. The authors should state that if so.
> > > > >
> > > > > **Response**
> > > > >
> > > > > To clarify, in generative tasks (e.g., music/audio/speech separation, speech enhancement, and text-to-speech), we use a frozen decoder only to convert generated tokens back into audio. The decoder is not trained jointly with the tokenizer, nor is its performance directly evaluated. This design isolates the contribution of the discrete representations themselves.We have clarified this point in the paper to avoid potential confusion.
> > > > >
> > > > > ---
> > > > >
> > > > > ### Comment 2
> > > > > > Discuss more dataset coverage and potential biases. Clarify that handling non-English, low-resource, or noisy-domain data would need further extensions.
> > > > >
> > > > >
> > > > > **Response**
> > > > > We agree that evaluating performance in low-resource, multilingual, and noisy settings is critical and that further extensions are needed to fully address these challenges. Our current benchmark already includes multilingual and low-resource scenarios through two ASR tasks in Welsh and Basque from CommonVoice 17.0. We deliberately selected these languages because they are not included in the training data of any evaluated tokenizer, ensuring a fair comparison. In addition, both datasets are inherently challenging due to their low-resource nature and the high variability introduced by crowd-sourced data collection, which includes noise and speaker diversity. This setup allows us to probe tokenizer robustness under realistic low-resource conditions, settings that remain challenging even for strong continuous baselines. We are actively working to expand DASB along these dimensions by incorporating additional datasets, tasks, and tokenizer families. Finally, we note that Table 5 (“Tokenizer Category Trade-offs”) already provides a qualitative comparison of acoustic, semantic, and hybrid tokenizers across multiple criteria, including reconstruction quality, downstream performance, and robustness in low-resource settings, offering an initial step toward understanding these trade-offs.
> > > > >
> > > > > ---
> > > > >
> > > > > ### Comment 3
> > > > > >Writing issues.
> > > > >
> > > > >
> > > > > **Response**
> > > > > We have applied the requested changes in the revised draft.

---

### Review · Reviewer_ygXm · 2025-11-09

**Summary Of Contributions:**

This paper introduces a new benchmark to evaluate the quality of audio representations over a variety of discriminative and generative tasks. Compared to prior works that mainly evaluates the quality based on reconstruction quality, this paper proposes to directly evaluate on various downstream tasks without the reconstruction step, as reconstruction may not accurately reflect the performance for the task of interest.

The authors show that semantic tokens generally outperform acoustic ones, and proposes other suggestions for training

Strengths:
1. The idea to evaluate the quality of tokenization for downstream tasks is interesting and very relevant in practice as the performance on downstream tasks is what people care about.
2. The experiments are extensive performed on a wide variety of tasks. The results provide clear overall picture about the strengths and disadvantages of existing tokenizers on different tasks.

Weakness:
1. The benchmark datasets are mainly from existing datasets in previous works and do not introduce new data sources. The tasks are also somewhat standard classification and generative tasks for speech and sound. Perhaps new and more challenging tasks may be needed to fully evaluate the quality of existing discrete or continuous representations and suggest more interesting future directions for the community.
2. The paper lacks qualitative evaluation of the tokenization strategies (e.g. cluster plots of representation) or case study, which are both very helpful to understand the quality of tokenization/representation.
3. A minor weakness is that the benchmark seems to be more focused on speech than sound and music: there much more tasks for speech than sound and music.

**Audience:**

Yes

**Audience Explanation:**

This paper should be interesting to researchers working on speech and audio processing, as well as those interested in incorporating audio inputs to multi-model large language models.

**Claims And Evidence:**

Yes

**Claims Explanation:**

The claims are supported by extensive experiments. Different tokenization strategies are tested on a wide variety of discriminative and generative tasks, providing strong empirical support for the claims.

**Requested Changes:**

1. The reviewer suggests that either design one or two new tasks and/or datasets that could better test the power of existing audio representations or provide a justification why the current choice of tasks and datasets is sufficient. This is critical for my recommendation.
2. The authors discover that the performance is not promising for sound and music tasks, and that discrete tokens are generally worse than continuous representations. It would be helpful for the community to provide some qualitative illustration/plots of the representations or case studies, which could provide more specific insights on how to improve these representations. This is less critical than the first point, but having this would strengthen the paper.

---

> ### Author Response · Authors · 2026-04-11
> **Requested Change**
>
> ### Comment 1
> > The reviewer suggests designing one or two new tasks and/or datasets to better test the power of existing audio representations, or providing a justification for why the current choice is sufficient. This is critical for my recommendation.
>
> **Response**
> Discrete audio tokens can be evaluated from multiple perspectives[1] . As discussed in the related work section, several benchmarks have been proposed to assess discrete audio representations. For example, Codec-SUPERB evaluates reconstruction quality using subjective scores and downstream performance on synthesized audio, while ESPnet-Codec and VERSA provide toolkits for training and evaluating codec models. However, these benchmarks rely on decoding the discrete tokens back into audio before evaluation. In contrast, DASB is the first benchmark specifically designed to evaluate the direct use of discrete audio tokens, without relying on waveform reconstruction. By removing the decoder, DASB isolates the information loss introduced by tokenization, allowing for a more accurate analysis of how discrete representations impact downstream tasks and multimodal language modeling.
>
> We first clarify that the goal of DASB is to evaluate the information loss introduced by discretization under a controlled and consistent setup. To this end, we use the same datasets and downstream architectures across tokenizers and ensure robustness through multi-seed evaluation, extensive hyperparameter tuning, and both shallow and deep downstream models. This design allows differences in performance to reflect properties of the representations, rather than training or modeling artifacts. Our benchmark covers a broad set of tasks that probe different aspects of audio representations. These include:
>
> - **Content-level tasks** (e.g., ASR, keyword spotting, intent classification), which require semantic understanding
> - **Acoustic-level tasks** (e.g., emotion recognition, speaker identification, speaker verification), which rely on fine-grained acoustic cues
> - **Generative tasks** (e.g., TTS, speech enhancement, separation), which test the ability to model and reconstruct signals
>
> In addition, we include **multilingual and low-resource settings** (Welsh and Basque from CommonVoice 17.0), where none of the evaluated tokenizers have seen training data, enabling evaluation under challenging conditions.
>
> We also include multilingual and low-resource settings (Welsh and Basque from CommonVoice 17.0), where none of the evaluated tokenizers have seen training data. This setup allows us to assess robustness under challenging conditions. This design follows prior benchmarks such as SUPERB[1], HEAR[2], and SpeechBench[3], but adapts them to discrete representations, which are more sensitive to bitrate, codebook size, and downstream model design. In addition to benchmarking, DASB provides analysis on these factors, offering guidance on tokenizer design and selection.
>
> [1] Yang, Shu-wen, et al. "Superb: Speech processing universal performance benchmark." arXiv preprint arXiv:2105.01051 (2021).
>
> [2] Turian, Joseph, et al. "Hear: Holistic evaluation of audio representations." NeurIPS 2021 Competitions and Demonstrations Track. PMLR, 2022.
>
> [3] Zaiem, Salah, et al. "Speech self-supervised representation benchmarking: Are we doing it right?." arXiv preprint arXiv:2306.00452 (2023).
>
> ---
>
> ### Comment 2
> > The authors discover weaker performance on sound and music tasks. It would be helpful to provide qualitative illustrations or case studies to better understand these limitations.
>
> **Response**
> Regarding non-speech domains, we acknowledge that the general observed performance of the tokenizers is weaker on sound and music tasks. Rather than being a limitation of the benchmark, we view this as an important finding: it highlights that current tokenizers are less effective outside of speech. To further investigate this, we have added three additional tasks:
>
> - **UrbanSound8K (event classification)**: 8732 labeled audio clips (≤ 4s) across 10 classes.
> - **NSynth (pitch classification)**: classifies instrumental sounds into 88 pitch classes.
> - **GTZAN (music vs. speech)**: binary classification with 120 tracks (60 per class), each 30 seconds long.
>
> All results are reported in Tables 4 and 11, and summarized in Figure 3. These additions strengthen coverage of sound and music, and confirm the observed trends.

---

### Review · Reviewer_GNDm · 2026-02-21

**Summary Of Contributions:**

This paper introduces DASB (Discrete Audio and Speech Benchmark), a comprehensive framework for evaluating discrete audio tokens across speech, general audio, and music domains. The benchmark systematically compares semantic, acoustic, and hybrid tokenizers on both discriminative tasks (ASR, speaker verification, emotion recognition, etc.) and generative tasks (speech enhancement, separation, TTS). The key contribution is the direct evaluation of discrete tokens without relying on decoder reconstruction, isolating the information loss introduced by quantization.

**Audience:**

Yes

**Audience Explanation:**

Discrete audio tokens are increasingly central to multimodal LLMs. Understanding their limitations is crucial for researchers building such systems. The finding that discrete tokens lose significant information compared to continuous features has direct implications for system design. The set of experiments to evaluate the understanding and generation tasks of different types of speech tokens is exhaustive, and the resulting take-away messages appear insightful.

**Claims And Evidence:**

Yes

**Claims Explanation:**

I found that a majority of claims made in this paper are well supported with empirical results. However, I am not entirely certain on the following perspectives:

- Regarding the performance difference between continuous and discrete representations, the authors claim that it comes from the information loss due to quantization in the discrete approaches. This should be justified more thoroughly. I wonder whether this hypothesis is true, as it would imply that simply increasing the bitrate could address the gap in the discrete approach, while this does not appear to be valid in practical cases.

- The authors claim that DASB directly evaluates tokenizers without a decoder. This claim confuses me because the generative tasks in Table 3 require a pre-trained decoder or vocoder.

- While the authors carefully design the experimental protocol for fair and rigorous benchmarking, the tokenizers differ in training data, sample rates, and architectures simultaneously, making causal attribution difficult.

**Requested Changes:**

It would be appreciated if you could address my comments above. Additionally, I have a couple of requests as listed below.

- It would be appreciated by readers who are not familiar with this field if the tables showing experimental results indicated whether models are equipped with acoustic or semantic tokenizers.
- The main tables are too dense, making it difficult for readers to grasp the global tendency. I suggest creating radar plots similar to Figure 3 but with more detail.
- While I understand that you adopt simple architectures for the downstream tasks for fair evaluations, providing some numbers with state-of-the-art level architectures would still be helpful for readers to know whether the scores are reasonable.
- What is #Q? I am fairly certain that this number is related to the bitrate (the number of codebooks?), but it is not defined in the paper. Additionally, the performance with respect to #Q differs depending on the methods. Could you elaborate more on this phenomenon?

---

> ### Author Response · Authors · 2026-04-11
> **Difference between continuous and discrete representations**
>
> We argue that the gap in performance between continuous and discrete representations is not explained by bitrate alone, but by the richness of the representation. If bitrate were the main factor, increasing it would be expected to improve the performance. However, we observe the opposite trend. As shown in Tables 2, 3, and 4, increasing the bitrate often degrades performance across tasks (see also Section 5.2). Higher bitrates do preserve more signal-level information and improve reconstruction quality. However, they also increase the number of tokens, which raises the input dimensionality and sequence length. This makes learning more difficult for downstream models. That is, our key point is how rich and structured the information is. A rich representation should support strong performance even with a simple downstream model. We observe this in continuous features (WavLM-large with layer aggregation), where a shallow head is sufficient to achieve good results. In contrast, discrete representations do not show the same behavior. Increasing bitrate does not improve performance, and often requires more complex downstream models to compensate for the added complexity and noise. This suggests that the additional tokens do not provide useful structure for learning and may introduce redundancy. To ensure a fair comparison, we use the same small and lightweight downstream head across all tokenizers. This setting isolates the effect of the representation itself, rather than model capacity.
>
> This observation is also supported by prior work. Recent studies show that higher bitrates do not always lead to better downstream performance. For example, Moshi [1] reports that lower bitrates can sometimes yield better results. Similarly, VoxtLM [2] shows that increasing the number of centroids from 200 to 1000 worsens CER (3.5 → 6.1) in TTS. These findings suggest that a higher bitrate can introduce redundancy and complexity that harm semantic learning, even if reconstruction improves. We have clarified this point in the revised paper(section 5.2 -Impact of Codebook Size and Bitrate):  the gap is better explained by the quality and usability of the representation, rather than the amount of information measured by bitrate.
>
>
> [1] Défossez, Alexandre, et al. "Moshi: a speech-text foundation model for real-time dialogue." arXiv preprint arXiv:2410.00037 (2024).
>
> [2]Maiti, Soumi, et al. "VoxtLM: Unified Decoder-Only Models for Consolidating Speech Recognition, Synthesis and Speech, Text Continuation Tasks." ICASSP 2024-2024 IEEE International Conference on Acoustics, Speech and Signal Processing (ICASSP). IEEE, 2024.

---

> ### Author Response · Authors · 2026-04-11
> **Confusion about decoder-free evaluation**
>
> We agree that this point may cause confusion, and we have added clarification in the revised draft (Page 4). DASB evaluates tokenizers at the representation level. All processing and downstream model training are performed directly on discrete tokens, not on reconstructed audio. In this sense, the decoder is not part of the evaluation pipeline. For generative tasks (Table 3), a pre-trained decoder or vocoder is used only at the final stage to convert generated tokens into waveform. The model itself operates entirely in the discrete token space during both training and inference. This setup is consistent with standard generative pipelines, where the vocoder is used only for waveform reconstruction, while all intermediate modeling is done on tokens or latent representations. Therefore, DASB isolates the effect of tokenization by ensuring that learning and inference happen in the discrete space. The decoder does not influence model training or representation learning, and is only used for final audio generation.

---

> > ### Author Response · Authors · 2026-04-11
> > **Difference in tokenizer training setting**
> >
> > While the authors carefully design the experimental protocol for fair and rigorous benchmarking, the tokenizers differ in training data, sample rates, and architectures simultaneously, making causal attribution difficult.
> >
> > We agree that controlling all factors (training data, sample rate, and architecture) would allow stronger causal conclusions. Our goal in DASB is different. We focus on evaluating existing tokenizers as they are used in practice. In real settings, tokenizers differ along multiple dimensions, and practitioners need guidance on which model to choose under these conditions.We therefore select a diverse set of tokenizers based on the following criteria:
> >
> >
> > (1) open-source availability with accessible checkpoints and code for reproducibility;
> >
> > (2) coverage of different quantization strategies (e.g., RVQ, SVQ, FSQ, K-means, and semantic distillation);
> >
> > (3) coverage of multiple domains, including speech, music, and general audio.
> >
> >
> > This design is similar to other benchmarks and leaderboards(e.g. SUPERB[1], HEAR[2], and SpeechBench[3]), which compare models under their standard configurations to provide practical insights.
> > We acknowledge that this setup does not isolate individual factors such as architecture or training data. Prior work has studied these aspects in more controlled settings (e.g., comparisons within a single architecture or dataset). In contrast, DASB aims to complement these studies by evaluating end-to-end tokenizer choices. Importantly, DASB goes beyond ranking. It provides analysis on key factors such as bitrate (number of codebooks), downstream model capacity, data scale, and initialization. These analyses help interpret performance trends and offer guidance on which tokenizers are best suited for specific tasks and conditions.
> >
> > [1] Yang, Shu-wen, et al. "Superb: Speech processing universal performance benchmark." arXiv preprint arXiv:2105.01051 (2021).
> >
> > [2] Turian, Joseph, et al. "Hear: Holistic evaluation of audio representations." NeurIPS 2021 Competitions and Demonstrations Track. PMLR, 2022.
> >
> > [3] Zaiem, Salah, et al. "Speech self-supervised representation benchmarking: Are we doing it right?." arXiv preprint arXiv:2306.00452 (2023).

---

> > > ### Author Response · Authors · 2026-04-11
> > > **Requested Change:**
> > >
> > > ### Comment 1
> > > > It would be appreciated by readers who are not familiar with this field if the tables showing experimental results indicated whether models are equipped with acoustic or semantic tokenizers.
> > >
> > > **Response**
> > > We have added a column in Table 1 to indicate whether each model uses an acoustic, semantic, or hybrid tokenizer.
> > >
> > > ---
> > >
> > > ### Comment 2
> > > > The main tables are too dense, making it difficult for readers to grasp the global tendency. I suggest creating radar plots similar to Figure 3 but with more detail.
> > >
> > > **Response**
> > > To improve readability, we have developed an interactive website that presents the results in a more accessible and structured way. Due to the anonymity requirement, we will release this website after the review process.
> > >
> > > ---
> > >
> > > ### Comment 3
> > > > While I understand that you adopt simple architectures for the downstream tasks for fair evaluations, providing some numbers with state-of-the-art level architectures would still be helpful for readers to know whether the scores are reasonable.
> > >
> > > **Response**
> > > To address this, we include both shallow and stronger downstream models in our evaluation. For each task, we report results with two types of architectures. For example, in ASR, we evaluate both a shallow 2-layer LSTM and a stronger Branchformer model. Similarly, for TTS, we include both a simple Transformer and a VALL-E-style architecture. The stronger models achieve performance comparable to reported results in the literature (see cited works), providing a reference to near state-of-the-art performance, while the shallow models ensure a fair comparison of representation quality across tokenizers.
> > >
> > > ---
> > >
> > > ### Comment 4
> > > > What is #Q? I am fairly certain that this number is related to the bitrate (the number of codebooks?), but it is not defined in the paper. Additionally, the performance with respect to #Q differs depending on the methods. Could you elaborate more on this phenomenon?
> > >
> > > **Response**
> > > Yes, #Q refers to the number of codebooks, which is directly related to the bitrate. We have added this definition in the caption for clarity. Regarding the observed differences across methods, we discuss this in Section 5.2. Increasing #Q increases the bitrate and preserves more signal-level information. However, it often degrades downstream performance. This happens for two main reasons. First, a larger number of codebooks produces more tokens, which increases sequence length and input dimensionality. This makes learning more difficult for downstream models. Second, not all codebooks contribute useful information. In RVQ-based tokenizers, early codebooks tend to capture structured features such as phonetics, while later codebooks often introduce redundancy or noise. This improves reconstruction quality but may hurt task performance. We also observe that higher #Q requires higher model capacity. For example, in ASR, a deeper model (Branchformer, Table 8) performs better than a shallow 2-layer LSTM (Table 2) at higher #Q. However, in low-resource settings, larger models are often not practical and may not train well.

---

### Decision · Action_Editor_gtxe · 2026-04-08

**Recommendation:** Accept with minor revision

**Additional Comments:**

The authors should provide a minor revision to address the reviewers' feedback, before the submission could be published.

**Audience:**

Yes

**Audience Explanation:**

All reviewers believe that some of TMLR's audience would be interested in the benchmark and the findings.

**Claims And Evidence:**

Yes

**Claims Explanation:**

Reviewers generally believed that the claims are supported by accurate, convincing and clear evidence. However, they also raised questions / requested changes in their reviews. The authors did not provide a response or a revision to address these questions. Nonetheless, two reviewers recommended acceptance assuming their requested minor edits will be incorporated into the final version. One reviewer, though believed that "majority of claims made in this paper are well supported with empirical results", found three key issues unresolved (due to lack of response): "(1) insufficient justification for attributing performance gaps to quantization, (2) inconsistency in the "decoder-free" claim, and (3) confounding factors in tokenizer comparisons."

---

> ### Author Response · Authors · 2026-04-11
>
> We would like to thank the AC for its decision and reviewers for their valuable feedback. We have provided a detailed answer to each reviewer. The summary of our answers to the raised issues is as follows.
>
> - **(i) Attribution of performance gaps to quantization.**
> We would like to clarify that the observed gap between continuous and discrete representations can not be explained by bitrate alone, but by the structure and usability of the representation. Our results (Tables 2–4, Section 5.2) show that increasing bitrate often degrades downstream performance, suggesting that additional tokens may introduce redundancy and learning complexity rather than useful information. We have strengthened this discussion in the revision.
>
> - **(ii) Inconsistency in the  “Decoder-free” evaluation.**
> We would like to clarify that DASB evaluates tokenizers at the representation level: all modeling and evaluation are performed in the discrete token space. For generative tasks, a pre-trained frozen decoder is used only at the final stage to reconstruct waveforms and is not part of the training or evaluation pipeline. This clarification has been added to the paper (section 2, page 3).
>
> - **(iii) Potential confounding factors in tokenizer comparison**
> We emphasize that DASB systematically controls and analyzes key factors. We provide consistent comparisons across tokenizers (Tables 2–4), study architectural choices such as codebook size (Fig.\~3), report computational trade-offs (Fig.~2), and include ablations (e.g., initialization and downstream head design). These analyses, along with discussions in Sections 5.1–5.2 and summaries in Table 5, help isolate contributing factors and explain observed trends.
>
> In the revised manuscript, all changes are highlighted in blue. We have added clarifications, addressed editorial issues, and expanded the benchmark by introducing three new tasks in the music and general audio domains. Overall, DASB is designed to identify where and why discrete representations lose information, providing actionable insights for practitioners and a foundation for future work. Full details are provided in the individual reviewer responses.

---

> > ### Comment · Action_Editor_gtxe · 2026-04-13
> >
> > Thank you, please go ahead and upload the camera ready version (without the highlights).